# Accelerating Block Coordinate Descent for LLM Finetuning via Landscape Correction

## Abstract

Training and finetuning large language models (LLMs) are resource-intensive tasks, with memory limitations being a key bottleneck. A classic optimization method, block coordinate descent (BCD), offers solutions by segmenting the trainable parameters into multiple blocks and optimizing one active block at a time while freezing the others, thereby significantly reducing memory cost. However, we identify that blindly applying BCD to train LLMs can be inefficient for two reasons. First, optimizing only the active block requires backpropagating through multiple deeper yet inactive blocks, resulting in wasteful computations. Second, the frozen blocks, when they are not quite close to optimality, can narrow the optimization landscape, potentially misguiding the training of the active block. To address these issues simultaneously, we propose integrating BCD with *landscape correction*, which unfreezes the inactive blocks and updates them in a cost-efficient manner during the same backpropagation as the update to the active block. We show that our method empirically improves vanilla BCD with minimal additional computation and memory. Experiments on 8B and 70B models demonstrate that our proposed method surpasses memory efficient baselines and matches Adam's downstream performance while reducing memory cost by $80\%$ compared to Adam.

## 1 Introduction

Large language models (LLMs) have gained significant popularity within the research community and industry. Training these models typically entails a pretraining phase on a vast dataset, followed by a series of finetuning adjustments to tailor the model for specific domain tasks. Both phases demand extensive computational resources, with memory being a primary constraint. For instance, optimizing a model containing $N$ billion parameters using standard mixed-precision training requires at least $18N$ gigabytes of GPU memory (refer to Section 2 for additional details). Given the usual limitations of GPU memory, this constraint impedes researchers from experimenting with larger models.

To address this practical challenge, researchers have developed memory efficient algorithms for LLM training such as parameter efficient finetuning (PEFT), including Adapter Houlsby et al. (2019), LoRA Hu et al. (2021), prompt tuning Lester et al. (2021), prefix tuning Li & Liang (2021), etc. These techniques focus on training a small set of additional parameters while maintaining the original pretrained model unchanged. Other memory efficient methods for full parameter training have also been investigated. For example, Galore Zhao et al. (2024) applies a low-rank space projection to both the gradient and the optimizer's states to reduce memory consumption.

In addition to the existing approaches, a classic optimization paradigm, known as *block coordinate descent* (BCD), holds a strong potential for memory efficient LLM training and finetuning. Intuitively, BCD reduces the memory cost by partitioning the trainable parameters into several blocks and optimizing over only one active block at a time. For instance, segmenting an $N$-billion parameter model into $D$ blocks decreases the memory consumption from $18N$ to $2N + \frac{16N}{D}$ GB, as only the gradient and optimizer states of the active block need to be stored. In fact, BCD has been the method of choice in many data science problems, with a wide array of variants developed for improving memory, performance, convergence, and efficiency; see, e.g., (Hsieh et al., 2008; Chang & Roth, 2011; Yu et al., 2012; Treister & Turek, 2014; Richtárik & Takáč, 2014).

In stark contrast to the previous memory efficient methods, and despite its intuitive memory benefits, BCD has been overlooked and rarely explored in the context of LLMs. It was not until very recently that Luo et al. (2024) proposed BAdam, which integrates BCD into an LLM finetuning framework by training each active block with several Adam steps. Even though BAdam has shown preliminary success in reducing memory cost during training and improving performance at test time, its direct use of vanilla BCD leaves at least two fundamental aspects to be questioned:

- Computing the (stochastic) gradient for a *single* active block via backpropagation necessitates calculating the partial derivatives of the activations of *multiple* deeper yet inactive layers. This is wasteful of computation as these partial derivatives are not even used to update their corresponding weights.
- Given that the training objective is highly nonconvex, and since all blocks are frozen except for the active one, BCD tends to be misled by *its local view of the optimization landscape*, which potentially slows down its convergence speed.

Standing on the ground offered by BAdam, we push the research frontier by proposing a simple remedy to the above two issues. Our method, termed BREAD, is a blend of two components: (1) Similarly to BAdam, we update the active block using several Adam steps (the BCD component); (2) differently, we unfreeze the inactive blocks and update them using lightweight memory efficient optimization techniques (the *landscape correction* component). Since landscape correction utilizes the gradients of the activations that are already calculated, it adds minimal additional computation and in fact addresses the wasteful computation issue. Furthermore, the landscape correction component provides BCD a better view of the optimization landscape for better updating the current active block, thereby addressing the second point of concern. In combination, BREAD maintains the memory efficient feature of BCD with improved learning capability and faster convergence. Our main contributions are outlined as follows:

- **Limitations of Standard BCD in LLMs:** Our research identifies two fundamental limits of vanilla BCD when applied to neural networks (hence LLMs): The wasted computation of gradients during backpropagation and the suboptimal landscape caused by freezing inactive blocks. These limitations partly explain why the application of BCD in neural networks is uncommon.
- **Blending BCD with Landscape Correction:** We propose a new method termed BREAD, which combines BCD with a landscape correction technique to address these two limitations simultaneously. It unfreezes some of (or all) the inactive blocks and updates them using memory efficient optimization techniques. BREAD maintains the memory efficiency of BCD with improved optimization ability.
- **Excellent Performance:** Our experiments on instruction tuning and preference optimization with the Llama 3.1-8B and Llama 3.1-70B models demonstrate that BREAD clearly outperforms state-of-the-art memory efficient training methods and achieves comparable downstream performance to that of Adam on five math benchmarks and MT-bench scores.

## 2 PRELIMINARIES ON BLOCK COORDINATE DESCENT FOR LLM TRAINING

Our main focus lies in improving the efficiency of BCD when utilized to finetune LLMs. Therefore, we first review some preliminary concepts of LLM and BCD in this section.

**Objective of training LLMs.** Consider minimizing a general objective function $\min_{\boldsymbol{W}} H(\boldsymbol{W}) = \frac{1}{n} \sum_{j=1}^{n} h_j(\boldsymbol{W})$, where $\boldsymbol{W} \in \mathbb{R}^d$, $n$ is the number of data samples. In the context of training/finetuning LLMs, $h_j(\boldsymbol{W})$ represents the negative log-likelihood of the autoregressive probability $\mathbb{P}_{\boldsymbol{W}}[\boldsymbol{y}_j|\boldsymbol{x}_j] = \prod_{s=1}^{m} \mathbb{P}_{\boldsymbol{W}}[\boldsymbol{y}_{j,s}|\boldsymbol{y}_{j,1:s-1}, \boldsymbol{x}_j]$ for the $j$-th prompt $\boldsymbol{x}_j$ and its corresponding $j$-th output $\boldsymbol{y}_j$. In most LLM models, this autoregressive probability is modeled by a transformer architecture Vaswani et al. (2017), and thus $\boldsymbol{W} \in \mathbb{R}^d$ encompasses all trainable parameters of the transformer, including the query, key, value, and output attention matrices, as well as the gate, up, and down projection matrices of each transformer layer.

**BCD for LLM training.** Instead of minimizing the objective function over the entire set of trainable parameters $\boldsymbol{W}$, the key idea of BCD is to break down this high dimensional optimization problem into a series of lower dimensional ones, thereby significantly reducing the memory requirement of GPU RAM. Specifically, BCD first splits the model parameters into $D$ block, i.e.,

$\boldsymbol{W} = \{\boldsymbol{W}_1, \cdots, \boldsymbol{W}_\ell, \cdots, \boldsymbol{W}_D\}$, where $\boldsymbol{W}_\ell \in \mathbb{R}^{d_\ell}$ and $\sum_{\ell=1}^{D} d_\ell = d$. The block partition in such a splitting can be very flexible. For instance, the block variable $\boldsymbol{W}_\ell$ can be either a single matrix or all the trainable matrices of a transformer layer. Then, at each block iteration, BCD updates only one active block while fixing the others at their most up-to-date values. This makes each sub-problem of BCD a $D\times$ smaller problem compared to the original one if the $D$ blocks are partitioned evenly. Suppose at the $(t+1)$-th block iteration the active block is $\boldsymbol{W}_\ell$, BCD solves the following problem:

$$\boldsymbol{W}_\ell^{t+1} \in \underset{\boldsymbol{W}_\ell \in \mathbb{R}^{d_\ell}}{\operatorname{argmin}} \frac{1}{|\mathcal{N}|} \sum_{j \in \mathcal{N}} h_j(\boldsymbol{W}_1^{t+1}, \cdots, \boldsymbol{W}_{\ell-1}^{t+1}, \boldsymbol{W}_\ell, \boldsymbol{W}_{\ell+1}^t, \cdots, \boldsymbol{W}_D^t) \tag{1}$$

where $\mathcal{N} \subseteq \{1, \cdots, n\}$ is a batch of the training dataset. Updating from block $\ell = 1$ to block $\ell = D$ is counted as one block-epoch. Since it is intractable to solve (1) exactly, one can instead approximate the solution by implementing $K$ Adam steps, as utilized in BAdam Luo et al. (2024).

**BCD is a memory efficient full parameter optimization method for LLM training.** It is evident to see that BCD is a full parameter optimization method, as all the trainable parameters $\boldsymbol{W}$ will be updated after one block-epoch. More importantly, BCD is also memory efficient.

Let us first analyze the memory consumption of the Adam optimizer under the mixed precision training setting Micikevicius et al. (2017). The memory cost is attributed to the storage of the model parameters, gradients, and optimizer states. We consider an LLM with $N$ billion parameters and express GPU memory consumption in gigabytes (GB). Initially, one must store the FP16 model parameters for the backpropagation (BP) process, requiring $2N$ memory. Additionally, the optimizer maintains a copy of the model in FP32 precision, consuming another $4N$ memory. The gradients, momentum, and second moment vectors are all stored in FP32 precision with each requiring $4N$ memory. Consequently, the total memory required is at least $18N$. For example, in order to train a Llama 3-8B or a Llama 3-70B model, Adam requires at least $144$ GB or $1260$ GB of GPU RAM, respectively, which can be prohibitive in limited memory scenarios.

In sharp contrast to Adam, BCD only requires storing the FP32 model parameters, gradients, and optimizer states for the *active block* $\boldsymbol{W}_\ell$, which is only $1/D$ of the memory consumption needed for all the parameters. Thus, in addition to maintaining an FP16 model that requires $2N$ memory, BCD needs a total of only $2N + \frac{16N}{D}$ memory. Therefore, for training a Llama 3-8B or a Llama 3-70B model and when $D = 32$ or $D = 80$ (partition each transformer layer as a block), BCD only needs roughly $20$ GB or $154$ GB of GPU RAM, respectively, which is significantly cheaper compared to the costs of Adam. For a more detailed analysis on memory cost, we refer to Luo et al. (2024).

## 3 LIMITATIONS OF BCD FOR NEURAL NETWORKS

Although BCD is proven to be memory efficient for training and finetuning LLMs, we will illustrate two major limitations of the BCD optimization scheme when it is used for training models induced by a neural network structure in this section. The results apply to the setting of training and finetuning LLMs as well. Motivated by these limitations, we will develop a more efficient training method based on BCD, which achieves superior performance compared to that of the vanilla BCD.

To ease our analysis, let us consider a $L$-layer feedforward neural network model:

$$\boldsymbol{z}_{\ell+1} = f_{\boldsymbol{W}_\ell}^\ell(\boldsymbol{z}_\ell), \ \forall 1 \le \ell \le L, \quad \text{with} \quad \boldsymbol{z}_1 = \boldsymbol{x}, \tag{2}$$

where $L$ is the total number of layers, $\boldsymbol{x}$ is the input, $f_{\boldsymbol{W}_\ell}^\ell$ is the $\ell$-th layer's transform.

**Limitation I: Ineffective utilization of intermediate derivatives during backpropagation.** Due to the compositional structure of deep neural networks, the gradients of the trainable parameters are calculated according to the chain rule. For example, taking the stochastic gradient of the $\ell$-th layer's parameters $\boldsymbol{W}_\ell$ requires computing the partial derivatives with respect to all the activation values of deeper layers, as shown in the following equation:

$$\frac{\partial H}{\partial \boldsymbol{W}_\ell} = \underbrace{\frac{\partial H}{\partial \boldsymbol{z}_{L+1}} \frac{\partial \boldsymbol{z}_{L+1}}{\partial \boldsymbol{z}_L} \cdots \frac{\partial \boldsymbol{z}_{\ell+2}}{\partial \boldsymbol{z}_{\ell+1}}}_{I_{\ell+1}} \frac{\partial \boldsymbol{z}_{\ell+1}}{\partial \boldsymbol{W}_\ell}, \tag{3}$$

where $H$ is the objective function of the neural network training problem. During the backpropagation process in optimization method like Adam, the intermediate partial derivatives $I_{\ell+1}$ of the

activations in (3) are properly utilized for computing the gradients of the $L, L-1, \cdots, \ell+1$-th layers' weight parameters as well. However, since BCD only updates the active block $\boldsymbol{W}_\ell$, the term $I_{\ell+1}$ is merely used for calculating the gradient of $\boldsymbol{W}_\ell$, resulting in ineffective utilization of the computed partial derivatives of the activations during backpropagation.

**Limitation II: Suboptimal landscape of BCD's sub-problem.** To tackle a training problem, optimization methods such as Adam optimize over all the trainable parameters $\boldsymbol{W}$, while the BCD optimization scheme (1) minimizes the objective over only the current active block, keeping the others fixed. Intuitively, BCD appears to narrow the optimization landscape of the training problem by freezing most of the parameters in each of its sub-problems, potentially eliminating better search directions that can lead to rapid decrease of the objective function. To establish such an intuition formally, we consider the following simple regression problem modeled by a 3-layer neural network:

$$\min_{\boldsymbol{W}_1, \boldsymbol{W}_2, \boldsymbol{W}_3} H(\boldsymbol{W}_1, \boldsymbol{W}_2, \boldsymbol{W}_3) := \|\boldsymbol{y} - \hat{\boldsymbol{y}}\|_2^2,$$

$$\text{where} \quad \boldsymbol{z}_2 = \sigma(\boldsymbol{W}_1 \boldsymbol{x}), \quad \boldsymbol{z}_3 = \sigma(\boldsymbol{W}_2 \boldsymbol{z}_2), \quad \hat{\boldsymbol{y}} = \boldsymbol{W}_3 \boldsymbol{z}_3.$$

Here, $\boldsymbol{x}$ is a non-zero input vector, $\boldsymbol{y}$ is the true label vector, and $\sigma(\boldsymbol{z}) := \max(0, \boldsymbol{z})$ is the ReLU activation function. The following proposition states that the optimization landscape of one of the BCD sub-problems is suboptimal in terms of minimizing $H$.

**Proposition 1.** *(suboptimal landscape of BCD's sub-problem) Define*

$$\widetilde{H}^* := \min_{\boldsymbol{W}_2} \|\boldsymbol{y} - \hat{\boldsymbol{y}}\|_2^2, \quad \text{and} \quad H^* := \min_{\boldsymbol{W}_1, \boldsymbol{W}_2, \boldsymbol{W}_3} \|\boldsymbol{y} - \hat{\boldsymbol{y}}\|_2^2.$$

*Here, $\widetilde{H}^*$ is the optimal value returned BCD that minimizes $H$ only over block $\boldsymbol{W}_2$, while fixing $\boldsymbol{W}_1$ and $\boldsymbol{W}_3$. Suppose that the fixed $\boldsymbol{W}_3$ in BCD has full column rank. If $\boldsymbol{z}_3^* := (\boldsymbol{W}_3^\top \boldsymbol{W}_3)^{-1} \boldsymbol{W}_3^\top \boldsymbol{y}$ has at least one negative entry, then $\widetilde{H}^* > H^*$.*

The proof of Proposition 1 is presented in Appendix C.2. We note that the full column rank assumption of the fixed weight matrix $\boldsymbol{W}_3$ is mild. In LLMs, the last matrix is typically the tall LM head matrix, which has more rows (vocabulary size) than columns (embedding dimension). In practice, a tall matrix often has full column rank. In addition, it is always feasible to have negative entries in $\boldsymbol{z}_3^*$ by choosing a specific $\boldsymbol{y}$. Moreover, although we prove such a result for a simple regression problem modeled by a 3-layer neural network, we expect it to occur more frequently in the training and finetuning of LLMs, as the training objective of LLMs is much more complicated. Finally, the result in Proposition 1 can be readily generalized to any nonnegative or bounded activation functions, e.g., Swish Ramachandran et al. (2017), SiLU Elfwing et al. (2018), Sigmoid, etc.

When fixing $\boldsymbol{W}_1$ and $\boldsymbol{W}_3$ and training only on the intermediate layer's weight matrix $\boldsymbol{W}_2$, the sub-problem of BCD represents a partial landscape of the full optimization problem. Proposition 1 demonstrates that this sub-problem might be incapable of finding the optimal value 0. It is important to note that this result does not necessarily imply that BCD cannot find an optimal solution. Recall that BCD is a full parameter optimization method, and the weight matrices $\boldsymbol{W}_1$ and $\boldsymbol{W}_3$ will indeed be updated in subsequent block iterations. Therefore, BCD may eventually converge to an optimal solution that achieves 0 function value. However, our analysis reveals that the sub-problem of BCD potentially excludes parts of the optimization landscape that provide search directions toward the optimal solution. This observation suggests that BCD might slow down the convergence speed compared to Adam.

Based on these limitations of BCD, we will design a new method to correct the landscape of BCD's sub-problem and simultaneously utilize the computed partial derivatives of the activations properly.

## 4 ACCELERATING BCD VIA LANDSCAPE CORRECTION

In this section, we propose a new BCD method with landscape correction to solve the two limitations revealed in Section 3 simultaneously.

### 4.1 THE BREAD METHOD

In Section 3, our analysis indicates that the incomplete landscape of BCD's sub-problem may slow down the convergence speed. To address this issue, one immediate approach is to apply Adam to

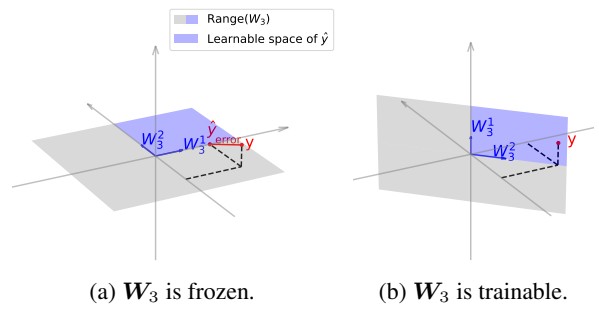

(a) $\boldsymbol{W}_3$ is frozen.      (b) $\boldsymbol{W}_3$ is trainable.

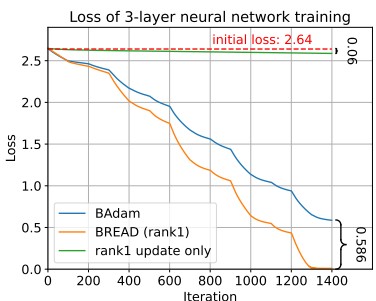

Figure 1: When $\boldsymbol{W}_3$ is frozen, the projection error $\|\boldsymbol{y} - \hat{\boldsymbol{y}}\|_2^2 > 0$. When $\boldsymbol{W}_3$ is trainable, the learnable space rotates to cover the label $\boldsymbol{y}$, the error $\|\boldsymbol{y} - \hat{\boldsymbol{y}}\|_2^2 = 0$.

Figure 2: With rank-1 landscape correction, BREAD converges significantly faster than BCD.

other blocks as well. However, this essentially reverts to using Adam and undermines the memory efficient property of the BCD optimization scheme. Fortunately, for the regression problem we analyzed in Section 3, we show in the following proposition that a *low-rank landscape correction* is sufficient to compensate for the incompleteness of BCD's sub-problem.

**Proposition 2** (rank-1 landscape correction)**.** *For any non-zero input $\boldsymbol{x}$ and label $\boldsymbol{y}$, there exists a rank-1 matrix $\boldsymbol{C}$ such that the following problem has optimal value 0:*

$$\min_{\boldsymbol{W}_2, \boldsymbol{C}} \|\boldsymbol{y} - \hat{\boldsymbol{y}}\|_2^2,$$

$$where \quad \boldsymbol{z}_2 = \sigma(\boldsymbol{W}_1 \boldsymbol{z}), \quad \boldsymbol{z}_3 = \sigma(\boldsymbol{W}_2 \boldsymbol{z}_2), \quad \hat{\boldsymbol{y}} = (\boldsymbol{W}_3 + \boldsymbol{C})\boldsymbol{z}_3.$$

The proof of Proposition 2 is shown in Appendix C.2. Motivated by this proposition, we propose to introduce additional *trainable low-rank correction matrices* to the matrices in the frozen inactive blocks $\{\boldsymbol{W}_{\ell'}\}_{\ell' \neq \ell}$, where $\boldsymbol{W}_\ell$ is the current active block. For simplicity, let us assume that each block is a matrix, and our design applies to general layer-wise or other types of partitions as well. Mathematically, all the inactive block matrices are corrected by low-rank correction matrices as follows:

$$\boldsymbol{W}_{\ell'} + \boldsymbol{C}_{\ell'} \text{ with rank}(\boldsymbol{C}_{\ell'}) \leq r, \quad \forall \ell' \neq \ell. \tag{4}$$

To maintain memory efficiency, we use the Burer-Monteiro factorization representation of a low-rank matrix Burer & Monteiro (2003):

$$\boldsymbol{C} = \boldsymbol{U}\boldsymbol{V}, \quad \boldsymbol{U} \in \mathbb{R}^{m \times r}, \boldsymbol{V} \in \mathbb{R}^{r \times n}. \tag{5}$$

Following the spirit of adapter Houlsby et al. (2019) and LoRA Hu et al. (2021), we only store the low dimensional matrices $\boldsymbol{U}$ and $\boldsymbol{V}$ rather than $\boldsymbol{C}$. This approach allows us to store the gradients and optimizer states for these much smaller sized matrices, resulting in only negligible additional memory consumption. We initialize $\boldsymbol{U}$ to zero and initialize $\boldsymbol{V}$ from uniform distribution; see Appendix A for detailed setup.

We note that one can add the landscape correction matrix $\boldsymbol{C}$ to each matrix in the inactive blocks or to part of the matrices, leading to two different variants. Additionally, we may also add a *full-rank* correction matrix $\boldsymbol{C}$, training it using an on-the-fly SGD method Lv et al. (2023) to maintain the memory efficient feature of BCD. We refer to Section 4.3 for details.

**Algorithm design.** Based on the above developments, we propose accelerating **b**lock coo**r**dinate d**e**scent via l**and**scape correction (BREAD). We present the detailed procedure in Algorithm 1. Suppose we can add a total of $P$ landscape correction matrices. BREAD first splits the model into $D$ blocks, which can be partitioned either in a layer-wise or matrix-wise manner. Then, each block sub-problem is approximately solved using $K$ steps of landscape corrected updates. For each landscape corrected update (Algorithm 1, line $9 - 16$), we sample a batch of data in a random reshuffled manner, and calculate the gradient of both active block and the correction matrices in one backward pass. Then, we update the active block and correction matrices with a single Adam step. Notably, optimizer states of the correction matrices are accumulated throughout the entire algorithm execution, as they occupy only negligible memory space.

---

**Algorithm 1:** BREAD: Block coordinate descent with landscape correction.

1 **input:** model parameters $\{\boldsymbol{W}_\ell^0\}_{\ell=1}^L$, number of blocks $D$, iterations per block $K$, training dataset: $\mathcal{D} = \{(x_j, y_j)\}_{j=1}^n$, batch size $B$.

2 **initialization:** block-epoch index $t \leftarrow 0$, the correction matrices $\boldsymbol{C}_j^0 \leftarrow \boldsymbol{0}$, $\forall j \in [P]$, and the corresponding optimizer states $\tilde{\boldsymbol{s}}_j^0 \leftarrow \boldsymbol{0}$, $\forall j \in [P]$.

3 **while** *stopping criterion not meet* **do**

4      generate a block partition $\pi = \{\pi_1, \ldots, \pi_D\}$ ;

5      **repeat** for one *block-epoch* $i \leftarrow 1, \ldots, D$

6          select correction matrices' indices $J \subset [P]$ as in (7);

7          $\boldsymbol{s}_{\pi_i}^{t,0} \leftarrow \boldsymbol{0}$ ;    // Re-initialize optimizer states for the active block

8          $\boldsymbol{W}_{\pi_i}^{t,0} \leftarrow \boldsymbol{W}_{\pi_i}^t$; $\tilde{\boldsymbol{s}}_J^{t,0} \leftarrow \tilde{\boldsymbol{s}}_J^t$ ;

9          **repeat** for *landscape corrected block updates* $k \leftarrow 1, \ldots, K$

10              sample a data batch in random reshuffled manner $\mathcal{D}_B = \{(x_j, y_j)\}_{j=1}^B \sim \mathcal{D}$;

11              **within** one backward pass on the data batch $\mathcal{D}_B$

12                 calculate the active block's grad. $\boldsymbol{g}_i^{t,k}$ and correction matrices' grad. $\tilde{\boldsymbol{g}}_J^{t,k}$;

13              **end**

14

             // Update the active block and correction matrices

15              $\boldsymbol{W}_{\pi_i}^{t,k}, \boldsymbol{s}_{\pi_i}^{t,k} \leftarrow \mathsf{AdamStep}(W_{\pi_i}^{t,k-1}, \boldsymbol{g}_{\pi_i}^{t,k}, \boldsymbol{s}_{\pi_i}^{t,k-1})$;

16              $\boldsymbol{C}_J^{t,k}, \tilde{\boldsymbol{s}}_J^{t,k} \leftarrow \mathsf{AdamStep}(C_{\pi_i}^{t,k-1}, \tilde{\boldsymbol{g}}_J^{t,k}, \tilde{\boldsymbol{s}}_J^{t,k-1})$;

17          **end**

18          $\boldsymbol{W}_{\pi_i}^{t+1} \leftarrow \boldsymbol{W}_{\pi_i}^{t,K}$; $\boldsymbol{C}_J^{t+1} \leftarrow \boldsymbol{C}_J^{t,K}$; $\tilde{\boldsymbol{s}}_J^{t+1} \leftarrow \tilde{\boldsymbol{s}}_J^{t,K}$; $\boldsymbol{s}_{\pi_i}^{t,K} \leftarrow \textbf{None}$;

19      **end**

20      $t \leftarrow t + 1$;

21 **end**

22 **return** *parameters* $\{\boldsymbol{W}_\ell^t\}_{\ell=1}^L$ *and correction matrices* $\{\boldsymbol{C}_j^t\}_{j=1}^P$.

---

We now carry out a simple experiment to validate the effectiveness of Algorithm 1 and the insights in Proposition 1 and Proposition 2. Specifically, we train a 3-layer neural network with rank-1 BREAD, with 100 Adam optimization steps for each sub-problem. As shown in Figure 2, rank-1 BREAD (orange) achieves significantly faster convergence than BCD (blue). The loss difference between rank-1 BREAD and BCD is eventually 0.586. This is a significant amount, as the loss is only decreased by 0.06, if we train $\boldsymbol{C}$ only (rank-1 update only, green). These empirical observations verify that the proposed combination of BCD and landscape correction accelerates the convergence of the individual scheme (BCD alone or landscape correction alone).

**Convergence result.** Below, we show that BREAD is a convergent method under some common assumptions; see Appendix C.1 for the detailed assumptions and analysis.

**Theorem 1.** *(informal) Under certain common assumptions, BREAD with deterministic gradient is a descent method with the following property*

$$H(\boldsymbol{W}^{t+1}) - H(\boldsymbol{W}^t) \leq -\mathcal{O}(\alpha K)\|\nabla H(\boldsymbol{W}^t)\|^2.$$

Consequently, BREAD finds $\varepsilon-$approximate stationary point with $\mathcal{O}(\varepsilon^{-2})$ iterations.

### 4.2 MEMORY AND COMPUTATIONAL EFFICIENCY OF BREAD

In this section, we analyze the memory and computational cost of BREAD, showing that BREAD only introduces marginal additional costs compared to vanilla BCD.

**Memory cost analysis.** To simplify the analysis, we consider a $D$-layer neural network where each layer consists of one matrix with dimensions $\mathbb{R}^{m \times m}$. The rank $r$ correction matrix introduces additional $2Dmr$ parameters. Since the correction matrix is primarily used for coarse-grained landscape correction, the rank is set to be small, e.g., $r \in [1, 8]$, making the additional memory required almost

negligible. In the scenario where $r = 4$, $D = 32$, and $m = 4096$, BREAD only increases memory cost by $\frac{Dr(m+m)}{m^2} \approx 2.6\%$ compared to BCD.

**Computational cost analysis.** We now show that the additional backward cost is also cheap, since the intermediate partial derivatives used for computing the active block's gradient can be directly used for computing correction matrices' gradients, as we have identified in (3). Specifically, the gradient of the correction matrix $C_j$ can be expressed as

$$\frac{\partial H}{\partial C_j} = \underbrace{\frac{\partial H}{\partial z_{L+1}} \frac{\partial z_{L+1}}{\partial z_L} \cdots \frac{\partial z_{j+2}}{\partial z_{j+1}}}_{\text{Computed in (3), } \forall j \geq \ell} \frac{\partial z_{j+1}}{\partial C_j}. \tag{6}$$

Clearly, when $j \geq \ell$, computing the (stochastic) gradient of $C_j$ only requries additional computation of $\frac{\partial (z_{j+1})}{\partial C_j}$, which is cheap given the low dimensionality after low-rank factorization representation, i.e., $C_j = U_j V_j$. We empirically measure the memory and epoch training time in Table 1.

### 4.3 PRACTICAL VARIANTS OF BREAD

**A computational efficient variant.** For simplicity, we consider an $L$-layer neural network where each layer consists of one weight matrix, and our block partition is layer-wise. Based on the derivation of (6), evaluating the gradients of the correction matrices is inexpensive for layers $\ell + 1, \ldots, L$. However, the gradient evaluation for layers $1, \ldots, \ell - 1$ is more costly, as it requires calculating $\frac{\partial z_{j+1}}{\partial z_j}$ for $j = 1, \ldots, \ell - 1$. These intermediate partial derivatives of the activations are not computed during the backpropagation to the active layer $\ell$. Therefore, one computationally efficient variant of BREAD is to add correction matrices only for layers $\ell + 1, \ldots, L$. This leads to two strategies of selecting correction matrices:

$$J = \begin{cases} [P], & \text{if use BREAD} \\ \{j \mid j \in [P], \; C_j \text{ corrects layers } \ell + 1, \ldots, L\}, & \text{if use BREAD-partial} \end{cases} \tag{7}$$

**A full-rank memory efficient variant.** The previous implementation uses low-rank matrices for landscape correction. Alternatively, one can apply a potentially full-rank linear transformation to correct features. This can be achieved through a memory efficient on-the-fly SGD update on the inactive blocks. Specifically, due to the compositional structure of neural networks, the gradient of the model is computed from the deep layers to the shallow layers. The strategy is to perform an SGD update on a matrix whenever its (stochastic) gradient is available, and then immediately discard the corresponding gradient after the update. We term this approach BREAD-SGD. It introduces additional memory cost for storing the gradient of the largest matrix, but this overhead is usually negligible. We formally present this variant in Algorithm 2.

We compare the performance of these two variants of BREAD in Section 5.4.

## 5 NUMERICAL EXPERIMENTS

We evaluate the proposed BREAD in finetuning Llama 3.1-8B and Llama 3.1-70B model on math finetuning and instruction tuning tasks, comparing its memory cost, time cost and downstream performance with full training algorithm and memory efficient baselines.

### 5.1 SETUP

We begin by introducing the experimental setup.

**Baselines.** We compare BREAD with **1) BAdam** Luo et al. (2024), which applies vanilla BCD algorithm with Adam as the inner solver; **2) LoRA** Hu et al. (2021), which freezes the pre-trained weight and only updates the injected low-rank adapters; **3) Galore** Zhao et al. (2024), which projects the gradient into low-rank spaces for reducing the memory cost; **4) Adam** Kingma (2014), which serves as the full parameter training baseline.

**Math finetuning.** We finetune the Llama 3.1-70B and Llama 3.1-8B models on MathInstruct dataset Yue et al. (2023) for 3 epochs, which contains 260K questions that covers wide range of fields

| Model | Method | Peak Memory Cost | Epoch GPU Hour | GPU # |
|-------|--------|-----------------|----------------|-------|
| Llama 3.1-70B | Adam (estimated) | 1260 GB+ | – | 16+ A100-80G |
| | LoRA | 296.8 GB | 213.1 | 8 A100-40G |
| | BAdam | 276.2 GB | 119.0 | 8 A100-40G |
| | BREAD | 288.6 GB | 212.4 | 8 A100-40G |
| | BREAD-partial | 288.6 GB | 152.7 | 8 A100-40G |
| Llama 3.1-8B | Adam | 208.2 GB | 37.3 | 8 A100-40G |
| | Galore | 40.5 GB | 10.3 | 1 A100-80G |
| | LoRA | 25.0 GB | 6.0 | 1 A100-40G |
| | BAdam | 21.8 GB | 3.3 | 1 A100-40G |
| | BREAD | 23.2 GB | 5.8 | 1 A100-40G |
| | BREAD-partial | 23.2 GB | 4.0 | 1 A100-40G |

Table 1: Memory footprint and time cost for finetuning models on MathInstruct.

in mathematics. The finetuned models are evaluated on 4 in-domain mathematical benchmarks, i.e., GSM8K, MATH, NumGLUE, and AQuA Cobbe et al. (2021); Hendrycks et al. (2021); Mishra et al. (2022); Ling et al. (2017), and 1 out-of-domain mathematical benchmarks, i.e., SimulEq Koncel-Kedziorski et al. (2016). The evaluations are based on 0-shot prompt and 4-shot chain-of-thought prompt, respectively. Due to the limited computational resource, we do not include the Adam's results for 70B model. Since there is no model parallel implementation released for Galore by the finish of the manuscript, we are unable to report its 70B results as well.

**Instruction tuning.** We perform supervise finetuning on the Llama 3.1-8B model using Alpaca-GPT4 dataset Peng et al. (2023), which contains 52K questions and corresponding GPT-4 generated answers. The model is evaluated on MT-bench Zheng et al. (2023) for examining the model's instruction-following capability.

**Preference optimization.** After the instruction tuning, we further align the tuned model using direct preference optimization (DPO) Rafailov et al. (2024) on Ultrafeedback dataset Cui et al. (2023). To compare with the baseline optimization methods more comprehensively, we report the evaluation results of using Adam instruction tuned model as base model, and using the same optimization method for both phases.

All the experiments are run for 3 epochs. The reported scores are the best one among checkpoints at epoch 1, 2, 3. The detailed hyper-parameters are presented in Appendix A.

### 5.2 MEMORY AND TIME COST MEASURE

In Table 1, we empirically measure the peak memory cost and one epoch's time cost of finetuning models on the MathInstruct dataset. The GPU hour is calculated as the training time × GPU number.

We set the LoRA rank to 64 to keep its number of trainable parameters (0.83 billion) close to a single block of BREAD (0.86 billion). Compared with LoRA, the proposed BREAD consumes slightly lower memory and costs about the same amount of training time. The computational-efficient variant BREAD-partial requires significantly less training time than BREAD, and is comparable to BAdam.

**Remark.** The reported memory cost is higher than the theoretical value, especially for the 70B model's experiments which requires distributed training. This additional memory cost arises from storing activation values and computational buffers, e.g. the gradient buffer for performing reduce scatter operation. Furthermore, the training time may have slight fluctuations for different runs.

### 5.3 FINETUNING PERFORMANCE

**Math finetuning.** The evaluation results on math benchmarks are shown in Table 2. For the 8B model's finetuning, BREAD beats all the other 3 memory efficient baselines in both 0-shot and 4-shot average score. Under the 0-shot setting, BREAD even outperforms Adam baseline by 1.1. For the finetuning of 70B model, BREAD outperforms BAdam in all tasks, demonstrating the effectiveness of landscape correction. Furthermore, BREAD beats LoRA in 8 out of 10 tasks.

| Base model: Llama 3.1-8B | | | | | | | | | | | |
|---|---|---|---|---|---|---|---|---|---|---|---|
| **Method** | GSM8K | | MATH | | NumGLUE | | SimulEq | | AQuA | | **Avg.** | |
| | 0-shot | 4-shot | 0-shot | 4-shot | 0-shot | 4-shot | 0-shot | 4-shot | 0-shot | 4-shot | 0-shot | 4-shot |
| Base model | 17.8 | 52.5 | 8.6 | 23.2 | 25.7 | 40.6 | 12.2 | 28.8 | 19.3 | 43.7 | 16.7 | 37.8 |
| Adam | 62.3 | 64.9 | 17.4 | 22.9 | 56.4 | 56.8 | 28.6 | 33.5 | 44.9 | 52.8 | 41.9 | 46.2 |
| Galore | 46.7 | 57.2 | 16.2 | 22.9 | 42.8 | 45.0 | 28.7 | **32.3** | 47.8 | 48.4 | 36.4 | 41.2 |
| LoRA | 48.7 | 58.1 | 13.7 | 23.0 | 34.6 | 54.4 | 29.6 | 29.0 | 47.3 | **50.3** | 34.8 | 43.0 |
| BAdam | 53.9 | **58.3** | 17.2 | 23.6 | 53.7 | 57.2 | 32.5 | 32.8 | **50.4** | 49.6 | 41.5 | 44.3 |
| **BREAD** | **57.0** | 57.6 | **20.0** | **23.7** | **55.9** | **58.2** | **32.5** | 32.8 | 49.6 | 50.0 | **43.0** | **44.5** |
| Base model: Llama 3.1-70B | | | | | | | | | | | |
| **Method** | GSM8K | | MATH | | NumGLUE | | SimulEq | | AQuA | | **Avg.** | |
| | 0-shot | 4-shot | 0-shot | 4-shot | 0-shot | 4-shot | 0-shot | 4-shot | 0-shot | 4-shot | 0-shot | 4-shot |
| Base model | 58.8 | 79.4 | 24.9 | 41.4 | 43.7 | 55.8 | 26.3 | 38.1 | 52.0 | 64.2 | 41.1 | 51.2 |
| LoRA | **83.8** | 82.0 | **41.7** | 44.2 | 70.4 | 69.0 | 40.3 | 48.8 | 61.4 | 65.8 | 59.5 | 62.0 |
| BAdam | 81.4 | 82.9 | 40.3 | 43.8 | 68.1 | 69.7 | 50.0 | 52.7 | 65.3 | 70.1 | 61.0 | 63.8 |
| **BREAD** | 83.4 | **84.2** | 41.4 | **44.7** | **73.1** | **74.4** | **51.3** | **56.8** | **68.3** | **70.5** | **63.5** | **66.1** |

Table 2: Math evaluation results for models finetuned on MathInstruct dataset.

| **Method** | SFT | | DPO | |
|---|---|---|---|---|
| | GPT-4 | GPT-4o | GPT-4 | GPT-4o |
| Base model | 6.07 | 5.18 | 6.63 | 5.66 |
| Adam | 6.63 | 5.66 | 7.83 | 6.18 |
| LoRA | 6.52 | 5.60 | 7.48 | 5.95 |
| Galore | 6.33 | 5.48 | 6.99 | 5.93 |
| BAdam | 6.53 | 5.57 | 7.63 | 6.14 |
| BREAD | **6.77** | **5.82** | 7.68 | **6.31** |

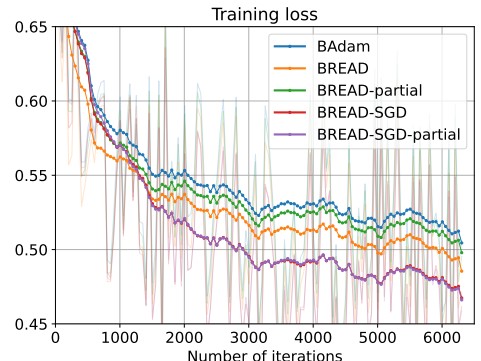

Table 3: MT-bench scores of different methods finetuning Llama 3.1-8B.

Figure 3: Convergence of BAdam, BREAD and its variants on MathInstruct dataset.

**Instruction tuning and DPO.** We report the MT-bench score evaluated by both GPT-4 and GPT-4o models in Figure 3. After SFT, the MT-bench score of all baseline approaches improves over the base model. BREAD achieves the highest scores in both evaluations, which are even higher than Adam, demonstrating the effectiveness of landscape correction. Based on the model finetuned by Adam, we further align the model using direct preference optimization (DPO). Notably, BREAD achieves the highest evaluation score by GPT-4o model.

## 5.4 ABLATION ON VARIANTS OF BREAD

We present 1-epoch training loss of BREAD and its variants in Figure 3. For reference, we also display the loss of BAdam. Here, BREAD-partial means that only deeper layers of the active block will be updated; the BREAD-SGD applies on-the-fly SGD update for all the frozen blocks; the BREAD-SGD-partial incorporates the feature of both, updating frozen blocks that are deeper than the active block using SGD. Since SGD usually requires higher learning rate, we scale the SGD's learning rate up by 10 times compared to the update of the active block. As we can see, all the variants converge faster than the baseline method BAdam, justifying the effectiveness of landscape correction. The BREAD outperforms BREAD-partial since it optimizes all the correction matrices in each iteration. The BREAD-SGD variants converge faster than the low-rank ones, which may attribute to the higher learning rate of the correction matrices and the high-rank update. Notably, the BREAD-SGD-partial exhibits similar convergence as BREAD-SGD.

### 5.5 ADDITIONAL EXPERIMENTAL RESULTS

We conduct additional ablation study on the hyperparameters of BREAD, and display the results in Appendix D. Below, we summarize the experimental results: **1) Effect of $K$.** We observe a consistent improved convergence speed when increasing $K$, which may due to that Adam can aggregate more historical information with larger $K$. **2) Effect of $r$.** BREAD with rank-2 correction significantly outperforms BAdam, and can be further accelerated with larger choices of $r$. **3) Effect of ordering strategies.** Different block ordering yields similar convergence behavior. **4) Convergence versus time**. In terms of wall-clock time, BREAD-SGD-partial yields the fastest convergence among the BREAD variants.

## 6 RELATED WORKS

**Block coordinate descent method.** Block coordinate descent (BCD) is a classic optimization paradigm that dates back at least to Hildreth (1957). It has gained popularity in recent years, due to its scalability and efficiency for many machine learning applications (Nesterov, 2012; Richtárik & Takáč, 2014; Peng & Vidal, 2023; Ding et al.; Peng & Yin, 2024). The community seems to converge to a consensus that, in order for BCD to be efficient, the problem it optimizes needs to possess the so-called coordinate-friendly structure (Shi et al., 2016). Nevertheless, deep networks are of a compositional nature and not coordinate-friendly, which is perhaps why recent surveys or books have never mentioned training deep networks as an application of BCD (Wright, 2015; Shi et al., 2016; Beck, 2017; Wright & Recht, 2022; Sayed, 2022). Recently, BAdam Luo et al. (2024) was proposed to finetune LLMs based on the BCD framework, where each block sub-problem is approximately solved using several Adam steps. Although BAdam achieved preliminary success, it is based on the vanilla BCD framework and shares the fundamental limitations we revealed in this work. In light of these, we believe identifying the limitations of BCD for LLM fintuning and fixing them entail certain insights, and this is what makes our contributions non-trivial and valuable.

**Memory efficient finetuning.** To address memory issue, multiple variants have been proposed. Parameter efficient finetuning (PEFT) methods achieve memory efficiency by only training small portion of (possibly extra) parameters while freezing most of the others, such as Adapter tuning Houlsby et al. (2019), prompt tuning, and prefix tuning Lester et al. (2021); Li & Liang (2021). Low-rank adaptation (LoRA) is perhaps the most popular technique that approximates model updates using two smaller, trainable low-rank matrices Hu et al. (2021). LoRA' variants have been proposed to address its rank constraints and further reducing the memory cost Lialin et al. (2024); Xia et al. (2024); Dettmers et al. (2023). Galore Zhao et al. (2024) projects the gradient into low-rank space so that it does not need to store the full gradient and optimizer states in the memory. LOMO updates parameters in real time during the backpropagation process Lv et al. (2023), so that one can perform SGD without store stochastic gradients. MeZO offers an alternative by approximating SGD using only forward passes Malladi et al. (2023), drawing from zeroth-order optimization that estimates stochastic gradients through the difference in function values. While this paper addresses the same application as these methods, they remain orthogonal to the proposed approaches. They can function as lightweight updates in the frozen layers for landscape correction.

## 7 CONCLUSIONS AND DISCUSSIONS

This paper investigates the application of a classic optimization method, known as BCD, to the finetuning of LLMs. We pinpoint two primary shortcomings of the standard BCD approach when applied to deep neural networks: the unnecessary computational overhead during backpropagation, and the misguiding optimization landscape caused by frozen blocks. To overcome these challenges, we introduce a new method termed BREAD, which unfreezes the inactive blocks and updates them in a lightweight manner. Our experimental results demonstrate that BREAD significantly enhances downstream task performance while maintaining the original BCD algorithm's memory and computational efficiency.

For future research, it would be intriguing to explore the potential of BCD in the (continual) pre-training phase of LLMs. Moreover, it is not necessary to apply feature correction at each iteration. Exploring the frequency of updating correction matrices is another meaningful direction, which can further save the computational cost.

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

## A    DETAILED EXPERIMENTAL SETUP

We introduce the detailed hyperparameters and experimental setup in this section.

**Global setup.** For all the experiments in math finetuning, instruction tuning and direct preference optimization, we fix the effective batch size to be 16 and train the model for 3 epochs. We use DeepSpeed ZeRO-3 to implement all the experiments that require distributed training (shown in Table 1). For all the experiments, we apply gradient checkpointing to reduce the memory cost for storing activation values. We use mixed-precision training with BFloat 16 as the low-precision datatype except for Galore, where we follow the setup in its paper, using pure BFloat 16 and 8-bit Adam optimizer for reducing the memory cost. Since the downstream tasks' performance of the language model have high variability, we grid search learning rate from {1e-6, 1e-5, 5e-5} with cosine learning rate schedule, and report the best result among the checkpoints at the end of epoch 1, 2, 3. The grid search is not extensive due to our limited computation resources. The implementation of BAdam, Galore, LoRA are based on LLama-Factory Zheng et al. (2024).

**Math finetuning.** We randomly select 100,000 samples from the MathInstruct dataset and finetune all the models using the same samples. The benchmarks scores are evaluated using the MAmmoTH's repository[1] (without using program-of-thought). The rank of correction matrices $U$ and $V$ for BREAD and BREAD-partial is set to 8. We initialize $U$ as zero, and initialize $V$ from the Kaiming uniform distribution He et al. (2015), i.e. $\left(-\mathcal{U}(\frac{\sqrt{6}}{r}), \mathcal{U}(\frac{\sqrt{6}}{r})\right)$. The rank of LoRA is set to 80 and 64 for finetuning Llama 3.1-8B and Llama 3.1-70B, respectively, so that the trainable parameter number of LoRA is close to that of one BAdam/BREAD's active block. We follow the conventional setup to set the LoRA scaling factor $\alpha = 2\times$ LoRA rank. We set Galore's rank to be 256, with the period of re-calculating the projection matrix being 256. We set $K = 100$ for BAdam and BREAD.

**Instruction tuning and direct preference optimization.** The evaluation of MT-bench score is based on FastChat Zheng et al. (2023) using both GPT-4 and GPT-4o API. The maximum sequence is set to 1024 and 2048 for the experiments on Alpaca-GPT4 and UltraFeedback, respectively. For BAdam and BREAD, we solve each block sub-problem for 128 steps, i.e. $K = 128$.

---

[1]https://github.com/TIGER-AI-Lab/MAmmoTH

## B  BREAD-SGD Algorithm

We introduce a variant of the BREAD algorithm, termed BREAD-SGD, which employs Adam for updating the active block and on-the-fly SGD for the inactive block. The detailed procedure is outlined in Algorithm 2. Analogous to BREAD, BREAD-SGD partitions the model into $D$ distinct blocks and combines the gradient computation and update steps into a singular operation. Specifically, gradients are calculated on a layer-by-layer basis; active layers are updated using Adam, while inactive layers undergo a single SGD step. Once an inactive layer is updated, its gradient is discarded to enhance memory efficiency.

---

**Algorithm 2:** BREAD-SGD

---

1 **input:** model parameters $\{\boldsymbol{W}_\ell^0\}_{\ell=1}^L$, number of blocks $D$, iterations per block $K$, step size of inactive blocks $\beta$.

2 **initialization:** block-epoch index $t \leftarrow 0$, and the corresponding optimizer states $\tilde{\boldsymbol{s}}_j^0 \leftarrow \boldsymbol{0}, \ \forall j \in [P]$.

3 **while** *stopping criterion not meet* **do**

4      generate a block partition $\pi = \{\pi_1, \ldots, \pi_D\}$ ;

5      **repeat** for one *block-epoch* $i \leftarrow 1, \ldots, D$

6          select correction matrices' indices $J \subset [P]$;

7          $\boldsymbol{s}_{\pi_i}^{t,0} \leftarrow \boldsymbol{0}$ ;          // Re-initialize Adam optimizer states

8          $\boldsymbol{W}_{\pi_i}^{t,0} \leftarrow \boldsymbol{W}_{\pi_i}^t$;

9          **repeat** for *landscape corrected block updates* $k \leftarrow 1, \ldots, K$

10              sample a data batch in random reshuffled manner $\mathcal{D}_B = \{(x_j, y_j)\}_{j=1}^B \sim \mathcal{D}$;

11              **within** one backward pass on the data batch $\mathcal{D}_B$

12                  // Update the active block

                 $\boldsymbol{g}_{\pi_i}^{t,k} \leftarrow \frac{\partial H}{\partial \boldsymbol{W}_{\pi_i}}$;

13                  $\boldsymbol{W}_{\pi_i}^{t,k}, \boldsymbol{s}_{\pi_i}^{t,k} \leftarrow \mathsf{AdamStep}(W_{\pi_i}^{t,k-1}, \boldsymbol{g}_{\pi_i}^{t,k}, \boldsymbol{s}_{\pi_i}^{t,k-1})$;

14

15                  // Correct inactive blocks

                 **for** $\ell \in J$ **do**

16                      $\boldsymbol{g}_\ell^{t,k-1} \leftarrow \frac{\partial H}{\partial \boldsymbol{W}_\ell}$;

17                      $\boldsymbol{W}_\ell^{t,k} \leftarrow \boldsymbol{W}_\ell^{t,k-1} - \beta \boldsymbol{g}_\ell^{t,k-1}$;

18                      $\boldsymbol{g}_\ell^{t,k-1} \leftarrow \textbf{None}$;

19                  **end**

20              **end**

21          **end**

22          $\boldsymbol{W}_{\pi_i}^{t+1} \leftarrow \boldsymbol{W}_{\pi_i}^{t,K}$;   $s_{\pi_i}^{t,K} \leftarrow \textbf{None}$;

23      **end**

24      $t \leftarrow t + 1$;

25 **end**

26 **return** *parameters* $\{\boldsymbol{W}_\ell^t\}_{\ell=1}^L$.

---

# C  CONVERGENCE RESULT AND ADDITIONAL PROOFS

## C.1  CONVERGENCE OF BREAD

In this section, we establish a preliminary convergence result for the proposed algorithms. For brevity of expression, we analyze the BREAD-SGD. The analysis of BREAD and BREAD-partial follows the similar strategy. We begin by introducing the assumptions.

**Assumption 1.** *The function $H(\boldsymbol{W})$ is $L$-smooth on $\boldsymbol{W}$ and $L_i$-smooth on block $\boldsymbol{W}_i$ for blocks $i = 1, \cdots, D$:*

$$\|\nabla_{\boldsymbol{W}} H(\boldsymbol{W}) - \nabla_{\boldsymbol{W}} H(\bar{\boldsymbol{W}})\| \leq L\|\boldsymbol{W} - \bar{\boldsymbol{W}}\|. \tag{8}$$

$$\left\| \frac{\partial H}{\partial \boldsymbol{W}_i}\bigg|_{\boldsymbol{W}_i^1} - \frac{\partial H}{\partial \boldsymbol{W}_i}\bigg|_{\boldsymbol{W}_i^2} \right\| \leq L_i\|\boldsymbol{W}_i^1 - \boldsymbol{W}_i^2\|, i = 1, \cdots, D. \tag{9}$$

The block-wise smoothness (9) can be naturally induced by the function smoothness (8).

**Assumption 2.** *There block derivatives $\nabla_{\boldsymbol{W}_i} H(\boldsymbol{W})$ are uniformly bounded by a constant $G$:*

$$\|\nabla_{\boldsymbol{W}_i} H(\boldsymbol{W})\| \leq G, i = 1, \cdots, D.$$

**Theorem 2** (Descent of BREAD). *Under Assumption 1 and Assumption 2, the Algorithm 2 with deterministic gradient achieves the following descent after each block-epoch of updates:*

$$H(\boldsymbol{W}^{t+1}) - H(\boldsymbol{W}^t) \leq -\mathcal{O}(\alpha K)\|\nabla H(\boldsymbol{W}^t)\|^2, \tag{10}$$

*under the step size choice $\alpha \leq \min\{\frac{\lambda}{2LK^2}, \frac{\lambda^2}{24LKG}, \frac{LG}{2}, \frac{108D}{16G^2K^3}\}$.*

By telescoping (10) from $t = 1, \cdots, T$, and divide each side by $T$, we obtain the sample complexity of $\|\nabla H(\boldsymbol{W}^t)\|^2 = \mathcal{O}(K/T)$. The proof of Theorem 2 is based on the following Lemma, which establishes the descent property of one block sub-problem.

**Lemma 1.** *Under Assumption 1 and Assumption 2, the Algorithm 2 update yields the following approximate descent property:*

$$H(\boldsymbol{W}_i^t) - H(\boldsymbol{W}_{i-1}^t) \leq -\frac{\alpha K}{4G}\|\nabla H(\boldsymbol{W}_{i-1}^t)\|_2^2$$

*Proof.* The proof of Lemma 1 follows the analysis framework of Luo et al. (2024), except that we need to analyze the descent property of the update in correction matrices. We define some notations for the ease of expression. At the beginning of block epoch $t$, block sub-problem $i$, let $\boldsymbol{W}_{j,i}^t$ be the parameter of block $j$, $\boldsymbol{W}_i^t$ be the full parameter. Based on the smoothness property in Assumption 1, we have

$$H(\boldsymbol{W}_i^t) - H(\boldsymbol{W}_{i-1}^t) \leq \left\langle \nabla H(\boldsymbol{W}_{i-1}^t), \boldsymbol{W}_i^t - \boldsymbol{W}_{i-1}^t \right\rangle + \frac{L}{2}\left\|\boldsymbol{W}_i^t - \boldsymbol{W}_{i-1}^t\right\|_2^2$$

$$= \underbrace{\sum_{j \in [D] \setminus i} \left\langle \nabla_j H(\boldsymbol{W}_{i-1}^t), \boldsymbol{W}_{j,i}^{t+1} - \boldsymbol{W}_{j,i-1}^{t+1} \right\rangle}_{I_1} + \underbrace{\sum_{j \in [D] \setminus i} \frac{L_j}{2}\|\boldsymbol{W}_{j,i}^{t+1} - \boldsymbol{W}_{j,i-1}^{t+1}\|_2^2}_{I_2}$$

$$+ \left\langle \nabla_i H(\boldsymbol{W}_{i-1}^t), \boldsymbol{W}_{i,i}^{t+1} - \boldsymbol{W}_{i,i-1}^{t+1} \right\rangle + \frac{L_i}{2}\|\boldsymbol{W}_{i,i}^{t+1} - \boldsymbol{W}_{i,i-1}^{t+1}\|_2^2 \tag{11}$$

The analysis of the last two terms directly follows the strategy in BAdam. We now analyze the first two terms. Let $\beta = c_\beta \alpha$ be the step size for the correction matrices. Define the gradient bias term of the block sub-problem as $\boldsymbol{e}_{j,i}^t := \frac{1}{K}\sum_{k=1}^K \nabla_j H(\boldsymbol{W}_{i-1}^{t,k}) - \nabla_j H(\boldsymbol{W}_{i-1}^t)$, where $\boldsymbol{W}_i^{t,k}$ is the parameter at block epoch $j$, sub-problem $i$, inner Adam step $k$, we have

$$I_1 = K\beta \sum_j \left\langle \nabla_j H(\boldsymbol{W}_{i-1}^t), -\nabla_j H(\boldsymbol{W}_{i-1}^t) + \boldsymbol{e}_{j,i}^t \right\rangle$$

$$\overset{(i)}{\leq} -\sum_j \frac{3}{4}K\beta\|\nabla_j H(\boldsymbol{W}_{i-1}^t)\|_2^2 + \sum_j 4\beta K\|\boldsymbol{e}_{j,i}^t\|_2^2$$

$$\overset{(ii)}{\leq} -\frac{3}{4}K\beta\|\nabla_j H(\boldsymbol{W}_{i-1}^t)\|_2^2 + 4\beta K \sum_j \left\| \frac{1}{K}\sum_{k=1}^K \nabla_j H(\boldsymbol{W}_{i-1}^{t,k}) - \nabla_j H(\boldsymbol{W}_{i-1}^t) \right\|_2^2$$

$$\overset{(iii)}{\leq} -\frac{3}{4}K\beta\|\nabla_j H(\boldsymbol{W}_{i-1}^t)\|_2^2 + 4\beta \sum_j \sum_{k=1}^K \left\| \nabla_j H(\boldsymbol{W}_{i-1}^{t,k}) - \nabla_j H(\boldsymbol{W}_{i-1}^t) \right\|_2^2$$

$$\overset{(iv)}{\leq} -\frac{3}{4}K\beta\|\nabla_j H(\boldsymbol{W}_{i-1}^t)\|_2^2 + 4\beta K \sum_j \sum_{k=1}^K L_j \left\| \boldsymbol{W}_{j,i-1}^{t,k} - \boldsymbol{W}_{j,i-1}^t \right\|_2^2$$

$$\overset{(v)}{\leq} -\frac{3}{4}K\beta\|\nabla_j H(\boldsymbol{W}_{i-1}^t)\|_2^2 + \frac{4\beta^3 G^2 K^4}{3}DL$$

$$\leq -\frac{1}{2}K\beta\|\nabla_j H(\boldsymbol{W}_{i-1}^t)\|_2^2 \tag{12}$$

where $(i)$ uses Young's inequality, $(ii)$ applies the definition of $e_{j,i}^t$, $(iii)$ uses the fact that $(\sum_{i=1}^K a_i)^2 \leq K a_i^2$, $(iv)$ is due to the Assumption 1, $(v)$ is due to

$$\sum_{k=1}^K \|\boldsymbol{W}_{j,i-1}^{t,k} - \boldsymbol{W}_{j,i-1}^t\|_2^2 \leq \sum_{k=1}^K \left( \sum_{m=1}^{k-1} \|W_{j,i-1}^{t,m+1} - W_{j,i-1}^{t,m}\| \right)^2$$

$$\leq \sum_{k=1}^K ((k-1)\beta G)^2$$

$$= \beta^2 G^2 \frac{(K-1)K(2K-1)}{6} \leq \frac{\beta^2 G^2 K^3}{3},$$

and the last inequality is due to $\beta \leq \sqrt{\frac{3D}{16K^3}}$, which is ensured by $\alpha \leq \frac{108D}{16G^2K^3}$.

Similar to the analysis above, we can bound $I_2$ based on Assumption 2:

$$I_2 = \sum_j \frac{L_j}{2}\beta^2 \left\| \sum_{k=1}^K \nabla_j H(\boldsymbol{W}_{i-1}^{t,k}) - \nabla_j H(\boldsymbol{W}_{i-1}^t) \right\|_2^2 \tag{13}$$

$$\leq \sum_j \frac{L_j}{2}\beta^2 K 4G^2 \tag{14}$$

$$\leq 2G^2 K L \beta^2 \leq \frac{1}{4}K\beta G^2, \tag{15}$$

where the first inequality is due to Assumption 2, the second inequality is based on the fact that $L > L_j, \forall j \in [D]$, and the last inequality is due to $\beta \leq L/8$. Combine (12), (15) and (11), and given that $\beta = \alpha/4G$, we have

$$H(\boldsymbol{W}_i^t) - H(\boldsymbol{W}_{i-1}^t) \leq -\frac{1}{4}K\beta\|\nabla H(\boldsymbol{W}_{i-1}^t)\|_2^2 + \langle \nabla_i H(\boldsymbol{W}_{i-1}^t), \boldsymbol{W}_{i,i}^{t+1} - \boldsymbol{W}_{i,i-1}^{t+1} \rangle$$

$$+ \frac{L_i}{2}\|\boldsymbol{W}_{i,i}^{t+1} - \boldsymbol{W}_{i,i-1}^{t+1}\|_2^2$$

$$\leq -\frac{\beta K}{2}\|\nabla H(\boldsymbol{W}_{i-1}^t)\|_2^2 - \frac{\alpha K}{8G}\|\nabla_i H(\boldsymbol{W}_{i-1}^t)\|_2^2$$

$$\leq -\frac{\alpha K}{8G}\|\nabla H(\boldsymbol{W}_{i-1}^t)\|_2^2 \tag{16}$$

where the second inequality follows (Luo et al., 2024)'s analysis. Specifically, by following exact the same argument in (Luo et al., 2024) Lemma D.6 and Lemma D.7, the last two terms can be decomposed into a descent term $-\mathcal{O}(\alpha\|\nabla H(\boldsymbol{W}_i^t)\|)$ plus an error term $\mathcal{O}(\alpha\|\tilde{e}_i^t\|^2)$ that represents the gradient bias, where the error $\|\tilde{e}_i^t\|$ can be shown to be in the order of $\mathcal{O}(\alpha\|\nabla_i H(\boldsymbol{W}_i^t)\|)$, which can be eliminated by controlling the step size $\alpha$.

*Proof of Theorem 2.* Since (16) establishs exact the same descent property as in Luo et al. (2024) Corollary D.8, one can follow the same argument as in BAdam's analysis to prove the Theorem 2.

$$\square$$

## C.2 PROOF OF PROPOSITIONS

*Proof of Proposition 1.* We first show that $H^* = 0$. One can construct $\boldsymbol{z}_3 = [1, 0, \cdots, 0]^\top$ and $\boldsymbol{W}_3 = [\boldsymbol{y}, \boldsymbol{0}, \cdots, \boldsymbol{0}]$. Note that such a choice of $\boldsymbol{z}_3$ is always achievable by choosing a specific $\boldsymbol{W}_2$. Hence, 0 function value can be attained by the constructed feasible point. This yields $H^* = 0$ after realizing that the objective function must be nonnegative. We illustrate this issue in Figure 1.

In BCD, $\boldsymbol{W}_1$ and $\boldsymbol{W}_3$ are fixed. We further assume that the fixed $\boldsymbol{W}_3$ has full column rank. We split our discussion into two cases. Case I: $\boldsymbol{y} \notin \text{range}(\boldsymbol{W}_3)$. We trivially have $\widetilde{H}^* > 0 = H^*$. Case II: $\boldsymbol{y} \in \text{range}(\boldsymbol{W}_3)$. In this case, $\boldsymbol{z}_3^* := (\boldsymbol{W}_3^\top \boldsymbol{W}_3)^{-1} \boldsymbol{W}_3^\top \boldsymbol{y}$ is the unique point that can achieve 0 function value. However, since $\boldsymbol{z}_3^*$ has at least one negative entry and $\boldsymbol{z}_3 \geq 0$ (due to the ReLU activation), we have $\|\boldsymbol{z}_3 - \boldsymbol{z}_3^*\|_2^2 > 0$. Therefore, we have $\|\boldsymbol{y} - \hat{\boldsymbol{y}}\|_2^2 = \|\boldsymbol{W}_3(\boldsymbol{z}_3^* - \boldsymbol{z}_3)\|_2^2 > 0 = H^*$, where the last inequality follows from the full column rankness of $\boldsymbol{W}_3$. $\square$

*Proof of Proposition 2.* We construct $\boldsymbol{z}_3 = \boldsymbol{e}_1 = [1, 0, \cdots, 0]^\top$. Let $\boldsymbol{C} = \left[\boldsymbol{y} - \boldsymbol{W}_3^{(1)}, 0, \cdots, 0\right]$, where $\boldsymbol{W}_3^{(1)}$ is the first column of $\boldsymbol{W}_3$. Then, we have $\|(\boldsymbol{W}_3 + \boldsymbol{C})\boldsymbol{z}_3 - \boldsymbol{y}\|_2^2 = \|\boldsymbol{C}\boldsymbol{e}_1 - (\boldsymbol{y} - \boldsymbol{W}_3\boldsymbol{e}_1)\|_2^2 = 0$. $\square$

## C.3 ANALYSIS OF MULTI-LAYER MODEL AND CROSS ENTROPY LOSS

In this section, we generalize the Proposition 1 to $L$-layer neural network model and cross entropy loss. The corresponding numerical verification are presented in Appendix D.4. Let us consider an $L$-layer model:

$$\boldsymbol{z}_1 = \sigma(\boldsymbol{W}_1 x)$$
$$\boldsymbol{z}_i = \sigma(\boldsymbol{W}_i \boldsymbol{z}_{i-1}), \quad i = 2, \cdots, L-1$$
$$\hat{\boldsymbol{y}} = \boldsymbol{W}_L \boldsymbol{z}_{L-1},$$

where $\sigma(\boldsymbol{x}) = \max(0, \boldsymbol{x})$ is the ReLU activation function and $\boldsymbol{z}_i \in \mathbb{R}^{d_i}$.

### C.3.1 SUBOPTIMALITY ANALYSIS FOR $L$-LAYER MODEL

Let us first consider the general regression loss $\|\hat{\boldsymbol{y}} - \boldsymbol{y}\|_2^2$, where $\boldsymbol{y}$ is the target we aim to fit.

**Effect of freezing $\boldsymbol{W}_L$.** When $\boldsymbol{W}_L$ is full column rank, the optimal $\boldsymbol{z}_{L-1}$ we seek to fit is the least square solution $\boldsymbol{z}_{L-1}^* = (\boldsymbol{W}_L^\top \boldsymbol{W}_L)^{-1} \boldsymbol{W}_L^\top \boldsymbol{y}$. When $\boldsymbol{z}_{L-1}^*$ contains negative entries, it cannot be fit due to the non-negativity of the ReLU function, which induces the suboptimality:

$$\min_{\boldsymbol{W}_1, \cdots \boldsymbol{W}_{L-1}} \|\hat{\boldsymbol{y}} - \boldsymbol{y}\|_2^2 > \min_{\boldsymbol{W}_1, \cdots, \boldsymbol{W}_L} \|\hat{\boldsymbol{y}} - \boldsymbol{y}\|_2^2.$$

**Effect of freezing intermediate layers.** Each intermediate layer performs the transformation $\boldsymbol{z}_i = \sigma(\boldsymbol{W}_i \boldsymbol{z}_{i-1}) := \mathcal{M}_i$. When $W_i$ is trainable, we have range$(\mathcal{M}_i) = \mathbb{R}^{d_i+}$ when $\boldsymbol{z}_{i-1} \neq 0$. However, when $\boldsymbol{W}_i$ is frozen, range$(\mathcal{M}_i)$ is limited to the projected "restricted" column space of $\boldsymbol{W}_i$, where "restricted" means that the column combination should be positive, due to the positivity of $\boldsymbol{z}_{i-1}$.

### C.3.2 SUBOPTIMALITY ANALYSIS FOR CROSS ENTROPY LOSS

Without loss of generality, let us assume that the ground truth label is the first class. The cross entropy loss is given by $-\log\left(\exp \hat{y}_1 / (\sum_{i=1}^{m} \exp \hat{y}_i)\right)$, where $m$ is the number of classes. Consider the case where the weight of the last layer $\boldsymbol{W}_L$ has a row with the same weight as the first row, i.e. $\exists j$ such that $\boldsymbol{W}_L^{(j)} = \boldsymbol{W}_L^{(1)}$, we have $\hat{y}_j = W_L^{(j)} \boldsymbol{z}_{L-1} = W_L^{(1)} \boldsymbol{z}_{L-1} = \hat{y}_1$. In this case, we will never be able to drive the loss down to $-\log \frac{1}{2}$:

$$-\log\left(\frac{\exp \hat{y}_1}{(\sum_{i=1}^{m} \exp \hat{y}_i)}\right) > -\log\left(\frac{\exp \hat{y}_1}{\exp \hat{y}_1 + \exp \hat{y}_i}\right) = -\log \frac{1}{2}.$$

While it is not common for $\boldsymbol{W}_L$ to have exactly two same rows, one can expect large error when there are rows that form small angle, i.e.

$$\frac{\left\langle \boldsymbol{w}_L^{(i)}, \boldsymbol{w}_L^{(j)} \right\rangle}{\|\boldsymbol{w}_L^{(i)}\| \|\boldsymbol{w}_L^{(j)}\|} \approx 1.$$

## D    ADDITIONAL EXPERIMENTS

### D.1    MORE ABLATION STUDY RESULTS

We present additional ablation studies on the hyperparameters of BREAD, including the block switch frequency $K$, the rank of correction matrices $r$, and the block ordering strategies.

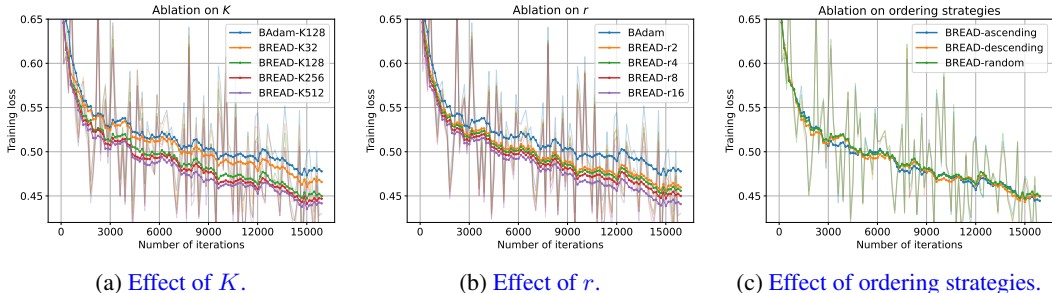

(a) Effect of $K$.    (b) Effect of $r$.    (c) Effect of ordering strategies.

Figure 4: Ablation study on the effect of Adam inner steps $K$, rank of correction matrices $r$, and block ordering strategies.

**Effect of $K$.** We present the effect of sub-problem update steps $K$ in Fig. 4a, which is by default 128 in our paper's experiments. The convergence of BAdam is provided for reference. Evidently, increasing $K$ consistently accelerates the convergence of BREAD for the examined range, where BREAD with $K = 512$ takes only half of the iterations to reach the final training loss of BREAD with $K = 32$. One possible explanation for the phenomenon is that when using larger $K$, the Adam update will aggregate more historical information in its momentum and second moment term, and thereby finds better search direction and scaling. We leave the scientific study of $K$ as a future direction. Notably, BREAD outperforms BAdam under all choices of $K$.

**Effect of $r$.** The effect of correction matrices' rank $r$ is shown in Fig. 4b, which is set to 8 in our paper. We note that by adding rank-2 correction matrices, BREAD converges significantly faster than BAdam, which corroborates our observation in Proposition 2. BREAD exhibits faster convergence as the rank increases, since larger rank offers higher freedom of search directions.

**Effect of ordering strategies.** We test the ordering strategies of ascending (from input layer to output layer), descending (from output layer to input layer), and random (select the layer in random reshuffling manner). As shown in Fig. 4c, different ordering strategy does not result in evident difference of convergence speed.

### D.2    CONVERGENCE IN TERMS OF TIME.

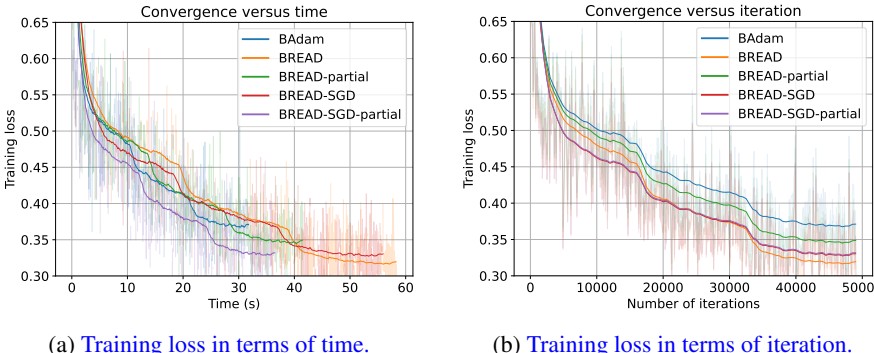

(a) Training loss in terms of time.    (b) Training loss in terms of iteration.

We finetune the Llama 3.1-8B model on MathInstruct dataset for 3 epochs, and report training loss convergence versus time/iteration in Appendix D.2. Notably, BREAD-SGD-partial achieves the fastest convergence in terms of time, and BREAD-SGD and BREAD-partial surpasses BAdam at certain points. All the BREAD variants achieve lower training loss than BAdam after 3 epochs.

### D.3 MAGNITUDE OF THE LEARNED PERTURBATION

Under the task of finetuning Llama 3.1-8B on MathInstruct, we plot the norm of the learned perturbation by BAdam and BREAD, where the magnitude is defined as the difference between the finetuned model and the pre-trained model: $\Delta W := W_T - W_0$. The results is shown in Fig. 6. Under the same step size choice, BREAD learns perturbation that has higher norm. This partially verifies that BREAD can expand the search space more efficiently than BAdam.

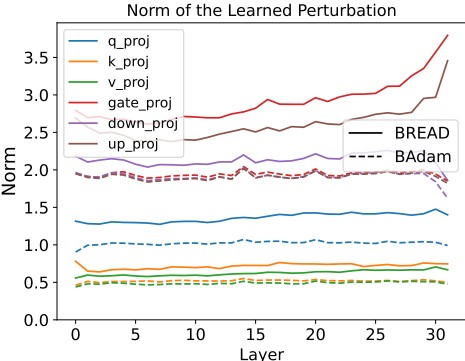

Figure 6: Norm of the learned perturbation. BREAD learns larger update than BAdam.

### D.4 FEATURE CORRECTION STUDY ON $L$-LAYER MODEL AND CROSS ENTROPY LOSS

In this section, we present the training loss of classification problem, where we use the cross entropy as the training objective. Specifically, we treat each layer as one block, and set the block switch frequency $K = 100$. We begin training with the first layer.

As shown in Fig. 7a, BREAD with rank-1 landscape correction converges dramatically faster than BAdam. In particular, BAdam only begins to converge rapidly after the 300 steps, when the final layer has been trained. This phenomenon supports our discussion in Appendix C.3.2, where we noted that a poorly trained final layer may hinder convergence. In Fig. 7b, we show that BREAD boosts the convergence for 8-layer neural network as well, which corroborates our $L$-layer analysis in Appendix C.3.1.

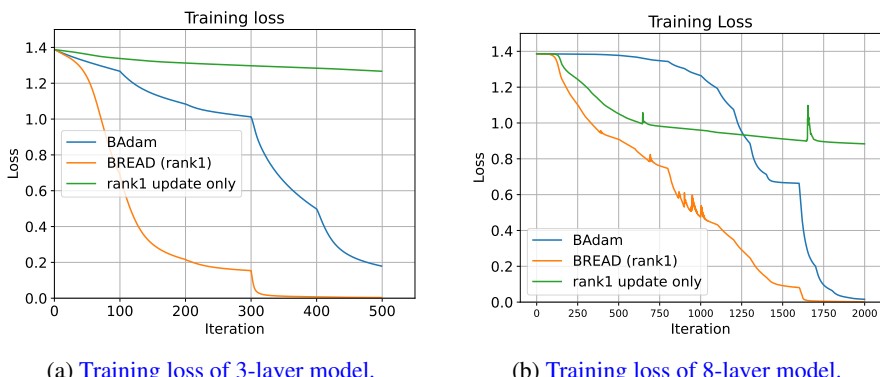

(a) Training loss of 3-layer model.

(b) Training loss of 8-layer model.

Figure 7: BREAD with rank-1 landscape correction converges dramatically faster than BCD for both 3-layer and 8-layer neural network training.

| Base model: Llama 3.1-8B | | | | | | | | | | | | |
|---|---|---|---|---|---|---|---|---|---|---|---|---|
| **Method** | GSM8K | | MATH | | NumGLUE | | SimulEq | | AQuA | | Avg. | |
| | 0-shot | 4-shot | 0-shot | 4-shot | 0-shot | 4-shot | 0-shot | 4-shot | 0-shot | 4-shot | 0-shot | 4-shot |
| Base model | 17.8 | 52.5 | 8.6 | 23.2 | 25.7 | 40.6 | 12.2 | 28.8 | 19.3 | 43.7 | 16.7 | 37.8 |
| Adam | 62.3 | 64.9 | 17.4 | 22.9 | 56.4 | 56.8 | 28.6 | 33.5 | 44.9 | 52.8 | 41.9 | 46.2 |
| Galore | 46.7 | 57.2 | 16.2 | 22.9 | 42.8 | 45.0 | 28.7 | 32.3 | 47.8 | 48.4 | 36.4 | 41.2 |
| LOMO | 31.9 | 53.9 | 15.6 | 22.7 | 39.5 | 39.8 | 22.6 | 28.2 | 36.2 | 44.9 | 29.2 | 37.9 |
| LoRA | 48.7 | 58.1 | 13.7 | 23.0 | 34.6 | 54.4 | 29.6 | 29.0 | 47.3 | **50.3** | 34.8 | 43.0 |
| BAdam | 53.9 | 58.3 | 17.2 | 23.6 | 53.7 | 57.2 | 32.5 | 32.8 | **50.4** | 49.6 | 41.5 | 44.3 |
| **BREAD** | 57.0 | 57.6 | **20.0** | **23.7** | **55.9** | **58.2** | 32.5 | 32.8 | 49.6 | 50.0 | **43.0** | 44.5 |
| **BREAD-partial** | 56.1 | **62.4** | 18.5 | 22.3 | 53.5 | 53.5 | 32.2 | **32.9** | 48.9 | 49.6 | 41.8 | 44.1 |
| **BREAD-SGD** | 56.9 | 60.6 | 19.6 | 21.4 | 54.1 | 58.2 | 31.5 | 31.8 | 48.0 | **50.8** | 42.0 | **44.6** |
| **BREAD-SGD-partial** | **58.0** | 60.1 | 17.8 | 20.8 | 53.9 | 54.3 | **32.5** | 31.5 | 47.3 | 47.0 | 41.9 | 42.7 |

Table 3: Math evaluation results for models finetuned on MathInstruct dataset.

| **0-shot results** | | | | | |
|---|---|---|---|---|---|
| Method | GSM8K | MATH | NumGLUE | SimulEq | AQuA | Avg. |
| Base model | 4.2 | 3.9 | 18.9 | 3.5 | 25.2 | 11.1 |
| BAdam | $10.9 \pm 1.27$ | $4.6 \pm 0.56$ | $24.4 \pm 3.00$ | $5.9 \pm 0.56$ | $28.3 \pm 1.50$ | 14.8 |
| BREAD | $\mathbf{14.0} \pm 1.19$ | $\mathbf{5.2} \pm 0.31$ | $28.1 \pm 1.31$ | $\mathbf{6.8} \pm 1.35$ | $28.1 \pm 2.46$ | **16.4** |
| **4-shot results** | | | | | |
| Method | GSM8K | MATH | NumGLUE | SimulEq | AQuA | Avg. |
| Base model | 6.2 | 6.3 | 20.2 | 5.6 | 28.3 | 13.3 |
| BAdam | $14.8 \pm 2.28$ | $5.9 \pm 0.39$ | $24.4 \pm 3.07$ | $5.9 \pm 1.19$ | $29.5 \pm 2.05$ | 16.1 |
| BREAD | $\mathbf{17.2} \pm 1.40$ | $\mathbf{6.0} \pm 0.29$ | $\mathbf{27.3} \pm 1.53$ | $\mathbf{5.9} \pm 1.18$ | $\mathbf{29.9} \pm 3.22$ | **17.3** |

Table 4: Math benchmark scores for Llama 3.2-1B finetuned on MathInstruct dataset. Results are averaged over 3 independent trials.

## D.5 ADDITIONAL EVALUATION ON MATH BENCHMARKS

In this section, we provide the math benchmark scores of BREAD variants in Table 3, as well as the averaged scores of finetuned Llama 3.2-1B models in Table 4. All the evaluation protocol follows the released code of MathInstruct.

As shown in Table 3, the variants with partial update have slight performance degrade compared with their full counterparts. Notably, All the BREAD variants outperform BAdam in 0-shot setting, and outperform other memory efficient baselines more significantly. The variants with partial update have slight performance degrade compared with their full counterparts.

**Llama 3.2-1B experiments.** We finetune Llama 3.2-1B on MathInstruct dataset for 3 independent trials, and report the averaged scores with standard deviation in Table 4. In both the 0-shot and 4-shot settings, BREAD outperforms BAdam across all tasks, supporting the effectiveness of the landscape correction.

