# OpenReview forum: "Accelerating Block Coordinate Descent for LLM Finetuning via Landscape Correction"
_ICLR.cc/2025/Conference — Submitted to ICLR 2025_

### Official Review · Reviewer_KSei · 2024-11-04

**Soundness:** 2
**Presentation:** 3
**Contribution:** 2
**Rating:** 5
**Confidence:** 3

**Summary:**

The paper presents  a method designed to improve the efficiency and convergence speed of block coordinate descent (BCD) in large language model (LLM) finetuning. The authors identify two main limitations of traditional BCD when applied to deep neural networks: (1) the ineffective use of computed gradients for inactive blocks during backpropagation, leading to wasted computational resources, and (2) the suboptimal optimization landscape created by freezing most blocks, which can hinder convergence.
To address these challenges, BREAD integrates a lightweight landscape correction mechanism that updates inactive blocks using low-rank matrix adjustments. So the method becomes SGD/ADAM on all layers but for inactive layers the LORA structure is constructed for SGD/ADAM to optimize.

**Strengths:**

The paper introduces an innovative combination of block coordinate descent (BCD) with a novel "landscape correction" approach, termed **BREAD**, tailored for large language model (LLM) finetuning. The originality stems from identifying limitations in the traditional BCD approach blending of BCD with lightweight, low-rank updates.
Empirical results from experiments on models like Llama 3.1-8B and Llama 3.1-70B demonstrate the method's effectiveness.
The paper is structured well.

**Weaknesses:**

1. In Case II of the proof for Proposition 1, it is unclear why \( z_3^* \) necessarily has at least one negative entry. The reasoning behind this is not evident.

2. The proposition does not adequately illustrate the global optimization landscape when applying BCD to a DNN. A single step in BCD resulting in a higher loss does not imply eventual failure. Therefore, the conclusion stating, 'our analysis reveals that the sub-problem of BCD potentially excludes parts of the optimization landscape that provide search directions toward the optimal solution,' is not clearly justified.

3. The described method involves performing SGD on all inactive layers for LoRA and also on active layers, making it more similar to Galore than true BCD. It is unclear how this approach achieves lower memory cost compared to Galore.

4. Demonstrating an extreme example in the propositions does not conclusively show that BCD performs worse than other methods in general. Experiments should be conducted to test the hypothesis that BCD restricts the search space. This could be verified by measuring the change in magnitude for each layer and comparing the results between BCD and BREAD to substantiate that BREAD indeed expands the search space more effectively.

**Questions:**

Please see Weakness.

---

> ### Author Response · Authors · 2024-11-21
> **Authors' Response**
>
> Thank you for your acknowledging the novelty of the BREAD and for your careful reading of the paper. We now address your concerns in a point-by-point manner. Major changes in the manuscript are highlighted in blue.
>
> **A. "Why $z_3^{*}$ has at least one negative entry?"**  We clarify that "$z_3^*$ contains negative entry" is an assumption instead of a fact, which holds with high probability. For any weight matrix $W$, let $y \in \mathbb{R}^{d}$ be sampled from Gaussian distribution $y \sim \mathcal{N}(0, 1)$, define $M:=(W^\top W)^{-1} W^{\top}$, the $i$-th entry of the least square solution $z^*$ is given by $z\_{i}^*=m\_i^\top y$, where $m_i$ is the $i$-th row of $M$. Since $y$ has symmetric distribution and $c_i$ is deterministic, $z_i^*$ will be a variable with symmetric distribution as well, indicating $\mathbb{P}(z_i^* > 0)=\frac{1}{2}$. Given the independence on different entries of $y$, we have $\mathbb{P}(\text{at least one} \  z_i^* < 0) = 1-\frac{1}{2^d}$. Hence, as $d$ increases, the probability approaches to 1 in an exponential way. The above analysis naturally generalize to $y$ sampled from any symmetric distribution.
>
> **B. "A single step in BCD resulting in a higher loss does not imply eventual failure."** We fully agree with your insightful observation. Hence, we did not argue that BCD will fail, as we have mentioned in the manuscript, line 197-199. However, the potential suboptimality issue identified in this paper may slow down the convergence of BCD. To this end, our objective is to accelerate BCD with low-cost landscape correction, given our observation on its potential suboptimality. We highlight that the empirical convergence and downstream performance indeed show that BREAD boosts BCD's performance by a large margin.
>
> **C. The described method is similar to Galore.** We clarify that BREAD is fundamentally different from Galore in terms of algorithmic design. Consequently, these two methods have different memory cost.
>
> *Fundamental difference in algorithmic design.* Galore aims to update all the parameters in each iteration. To reduce the memory cost, Galore projects the gradient into low-rank space to avoid storing the full gradient. In contrast, BREAD is a BCD-based method which mainly optimizes the active block, with "light adjustment" on the inactive blocks (either by low-rank correction or memory-efficient SGD correction).
>
> *Memory cost comparison.* We use a $L$-layer neural network model to illustrate the memory cost of BREAD and Galore. Consider a $L$-layer model with parameters $W_\ell \in \mathbb{R}^{m \times m}, \ell = 1, \cdots, L$. Let $r_1$ be the rank of BREAD correction matrices and $r_2$ be the projection rank of Galore. Notably, $r_1 \ll r_2$, since the correction matrices are verified to perform well under extreme low-rank; see Proposition 2. In our experiment, we let $r_1=8$ and follow Galore paper's setup to let $r_2=256$. Under the setting of $L=32$, $m=1024$, the number of stored variables for each method is presented below:
>
> * **Galore:** Model ($Lm^2$) + projected grad. \& optim.states ($3L mr_2$) = $L(m^2+3mr_2) \approx 5.87 \times 10^{8}$
> * **BREAD:** Model ($Lm^2$) + block grad. \& optim. states $(3m^2)$ + correction matrices grad. \& optim. states $(6Lmr_{1})$ = $L(m^2+6mr_{1}) + 3m^2 \approx 3.83 \times 10^{8}$.
> * **BREAD-SGD:** Model ($Lm^2$) + Block grad. \& optim. states $(3m^2)$ + grad. of one block $(m^2)$ = $(L+4)m^2 \approx 3.77 \times 10^{8}$.
>
> Evidently, BREAD stores less optimization variables in this setup. The memory cost of the paper's experiment follows the same analysis.
>
> **D. Verification on BCD's narrower search space.** Thank you for the insightful suggestion. We have plotted the norm of the difference between pre-trained model and finetuned model in Appendix D.3 of the latest manuscript. Indeed, BREAD learns perturbation with larger norm than BAdam, which is consistent across different layers and modules. In particular, BREAD learns perturbation with about $50%$ larger norm in down projection matrix. This partially justifies that BREAD can expand the search space more efficiently than BAdam.
>
> We believe our rebuttal has addressed all of your concerns with our best efforts. We hope it earns your trust in the quality of our work and your support for acceptance.

---

> ### Author Response · Authors · 2024-11-25
> **Looking Forward to Your Response**
>
> We hope our response addresses most of your concerns. In particular, we have shown $z_3^*$ will have at least one negative entry with high probability under mild assumptions on the weight $W_3$. We have clarified that BCD will not necessarily fail due to suboptimal landscape. Additionally, we have discussed the fundamental differences between Galore and BREAD, and explained the lower memory cost of BREAD using a neural network example. As requested by reviewer, we have conducted the experiment on verifying search space expansion of BREAD, where the results show that BREAD learns updates with larger magnitudes than BAdam under the same learning rate.
>
> As the deadline for the discussion phase approaches, we would greatly appreciate your feedback. Should you have any further questions or comments, please let us know so that we have sufficient time to address them. We would be happy to provide any additional clarifications or details.

---

> > ### Comment · Reviewer_KSei · 2024-11-28
> >
> > I appreciate the authors' feedback. However, I kept the concerns which is also raised by another reviewer regarding the limited technical contributions of BREAD. The techniques employed appear to be similar to those already used in related work. I will keep my score unchanged.

---

> > > ### Author Response · Authors · 2024-12-03
> > > **Response to Your Comment**
> > >
> > > Thank you for your reply. We hope that our response addresses most of your concerns.
> > >
> > > Regarding your concern on the novelty of the work, while BREAD adopts low-rank update or layer-wise SGD update for landscape correction in our manuscript, it is essentially a **general BCD algorithmic framework rather than tailored to LoRA or LOMO**. In fact, it naturally incoporates *any* lightweight memory efficient finetuning techniques while retaining the desirable features of BCD. Furthermore, the approach is **well-motivated by theoretical insight on BCD's suboptimality issue**, where inactive blocks that are far from optimality can interfere with the optimization of the active block. Additionally, we argue that **Substantially improved performance is also a non-trivial contribution.** For instance, from the technical perspective, the masked autoencoder is just an incremental combination of masking and autoencoder, yet it shines with strong performance. Hence, we believe BREAD is a novel and impactful framework, rather than a "A+B" combination.

---

### Official Review · Reviewer_3qN8 · 2024-11-04

**Soundness:** 2
**Presentation:** 2
**Contribution:** 2
**Rating:** 5
**Confidence:** 3

**Summary:**

The paper presents BREAD, a method designed to overcome limitations of block coordinate descent (BCD) in LLM fine-tuning by introducing landscape correction. The authors claim that BREAD unfreezes and updates inactive blocks using a lightweight optimization approach, addressing inefficiencies in derivative usage and suboptimal training landscapes. The method is evaluated through experiments on 8B and 70B LLMs, demonstrating memory savings and competitive downstream performance.

**Strengths:**

- The two issues of BCD are particularly intriguing and their demonstration via 3-layer NN is interesting.
- The experimental results look promising.

**Weaknesses:**

- **Presentation**. The paper’s presentation could be improved, particularly in terms of notations and the clarity of the method descriptions.
- **Limited technical contributions**. The technical novelty appears limited, as BREAD is essentially a combination of LoRA and BCD while BREAD-SGD is a combination of LOMO [1] and BCD.
- **Experiments**. Additional experiments, as outlined below, are necessary to strengthen the findings.

**Questions:**

1. **Algorithm 1**
    - Are there two separate instances of the Adam optimizer being used?
    - It is unclear how U and V are initialized and used.
    - What constitutes a single iteration? Is it defined as one while loop or one landscape correction update?
    - How is one epoch of training data sampled for Algorithm 1?
2. **Notation**
    - Ensure that all notations used in Section 2 are properly introduced.
    - What does $n$ represent in Section 4.2?
3. **Experimental settings**
    - How is alpha chosen for LoRA because rank 80 is a pretty weird setting.
    - The evaluation setup is not fully aligned with MathInstruct [2] regarding the choice of out-of-domain datasets and the number of shots used.
4. **Baselines**
    - How does the proposed method compare with LOMO?
    - The authors should include the performance of other variants for a more comprehensive evaluation.
5. **Ablation studies**. The authors should include ablation studies that explore the effect of the choice of D, K, and r on performance.
6. **Statistical significance**. The authors should report standard deviations across multiple random seeds to assess the robustness of the results.
- Minor:
    - In the proof of Proposition 2, $C = [y - W_3^{(1)}, 0, \ldots, 0]$ should be corrected.
    - Note that mathematical fine-tuning also falls under instruction tuning.
    - In Table 2 (Llama 3.1-8B, SimulEQ, 4-shot), the best result is incorrectly highlighted.

[1] Lv, Kai, et al. "Adalomo: Low-memory optimization with adaptive learning rate." *arXiv preprint arXiv:2310.10195* (2023).

[2] Yue, Xiang, et al. "Mammoth: Building math generalist models through hybrid instruction tuning." *arXiv preprint arXiv:2309.05653* (2023).

---

> ### Author Response · Authors · 2024-11-21
> **Authors' Response: Part I**
>
> Thank you for carefully reading the paper and for your recognition of our 3-layer neural network analysis. Our rebuttal takes your comments into full consideration, and we hope it will earn your support for acceptance.
>
> We now provide clarifications on the algorithmic design and address your concerns in a point-by-point manner. Major changes in the manuscript are highlighted in blue.
>
> **A. Questions on Algorithm 1, notation, and experiment settings.** We have improved the paper's presentation based on your suggestion; see Algorithm 1 and Section 4 of the latest manuscript. We now address your concern one by one:
> > "Are there two separate instances of the Adam optimizer being used?"
>
> We clarify that the Adam update of a parameter is purely determined by the gradient and the optimizer states. From the implementation perspective, we create a single Adam optimizer for both the active block and the correction matrices. The main difference is that the optimizer states (i.e. momentum and second moment) of active blocks will be re-initialized to zero each time when we switch block (Algorithm 1, line 7), while the optimizer states of the correction matrices are consistent and are not reset through the whole training procedure.
>
> > “It is unclear how U and V are initialized and used."
>
> We follow the LoRA's protocol to initialize $U$ to be zero and initialize $V$ from the Kaiming uniform, i.e. $\mathcal{U}(-\frac{6}{r}, \frac{6}{r})$. This ensures that for transformation $ y = U  V x$, the output $ y$ will have roughly the same variance as the input $ x$, which helps stabilize the training.
>
> > "What constitutes a single iteration? Is it defined as one while loop or one landscape correction update?"
>
> A single iteration in Figure 3 corresponds to one landscape correction update, i.e. one pass of line 10--15 of Algorithm 1.
>
> > "How is one epoch of training data sampled for Algorithm 1?"
>
> We sample one batch of data for each iteration; see Algorithm 1, line 10. The sampling strategy is based on random reshuffling, i.e. we shuffle the data at the beginning of each data epoch, ensuring each data point is seen exactly once per data epoch in a random order.
>
> > "What does $n$ represent in Section 4.2?"
>
> This is a typo and the $n$ should be corrected as $m$. We thank the reviewer for pointing it out.
>
> > "How is alpha chosen for LoRA because rank 80 is a pretty weird setting."
>
> For LoRA's rank, we follow the experimental setting in BAdam paper. The $\alpha$ is set to be $2 \times$ LoRA rank. The LoRA rank is set to match its trainable parameters with one active block of BREAD; see Appendix A, paragraph 3 for the detailed setup.
>
> > "The evaluation setup is not fully aligned with MathInstruct [2] regarding the choice of out-of-domain datasets and the number of shots used."
>
> Our evaluation is based on the official codebase released by MathInstruct. We note that it only contains 4 samples for some datasets, e.g. GSM8K, Aqua, and SAT. To align the shots of all benchmarks and keep the prompt samples consistent with the paper, we choose 4-shot as our few-shot evaluation. To exclude the effect of chain-of-thought prompt on model's performance, we report zero-shot score as well.

---

> ### Author Response · Authors · 2024-11-21
> **Authors' Response Part II**
>
> **B. Limited technical contributions: "BREAD is essentially a combination of LoRA and BCD while BREAD-SGD is a combination of LOMO and BCD".** We argue that BREAD is a well-motivated algorithmic framework, rather than trivial combinations of BCD and LoRA/LOMO. The framework naturally incoporates BCD with any light weight optimizers for alleviating suboptimality issue and accelerates the convergence of BCD. Specifically, the effectiveness of BREAD framework has been justified through theoretical insights, computational-efficient design, and competitive downstream performance under limited memory budget.
>
> *Theoretical insight.* In Proposition 1, we identified the suboptimal landscape issue of BCD, where optimizing a single block may lead to suboptimal solution. Proposition 2 further reveals that low-rank landscape correction can mitigate this suboptimality and improve the convergence of BCD. This insight is empirically validated in Figure 2 of the manuscript: while the low-rank update achieves only a negligible descent on its own, it significantly enhances the convergence of BCD. To our knowledge, this phenomenon has not been carefully examined in prior work. Furthermore, we have proved the faster convergence of BREAD compared with BAdam in Theorem 1 of Appendix C.
>
> *Computational-efficient algorithmic design.*  By leveraging the intermediate derivatives computed during the backpropagation of BCD, BREAD calculates the gradient of the correction matrices without requiring additional backpropagation steps. As a result, BREAD’s computational cost is significantly lower than Adam and comparable to LoRA; see Table~1 for detailed runtime comparisons. Based on the compositional feature of neural network, we further proposed BREAD-partial, a variant that incurs minimal computational overhead compared to BCD and is approximately 30\% faster than LoRA.
>
> *Competitive performance under limited memory budget.* Similar to LoRA and BAdam, BREAD is able to finetune Llama 3.1-70B model with only $4 \times$ A100-80GB GPUs. Compared with memory-efficient baselines, the model finetuned by BREAD attains the highest MT-bench score and average math benchmark score in instruction tuning tasks. Notably, BREAD outperforms Adam in supervised finetuning on Alpaca-GPT4, despite its $80%$ memory consumption.
>
> In summary, we would like to argue that BREAD is a well-motivated algorithmic framework with theoretical insights, computational-efficient design and competitive empirical performance, rather than a trivial combination of existing methods.
>
> **C. Additional experiments.** We conducted ablation study on sub-problem inner Adam step $K$ and rank of correction matrices $r$. Please see the Appendix D.1 of the revised manuscript. Below, we would like to summarize the results.
>
> *Effect of $K$.* Increasing $K$ consistently accelerates the convergence of BREAD for the examined range, where BREAD with $K=512$ takes only half of the iterations to reach the final training loss of BREAD with $K=32$. One possible explanation for the phenomenon is that when using larger $K$, the Adam update will aggregate more historical information in its momentum and second moment term, and thereby finds better search direction and scaling. Notably, BREAD outperforms BAdam under all choices of $K$.
>
> *Effect of $r$.* With rank-2 correction matrices, BREAD converges significantly faster than BAdam, which corroborates our observation in Proposition 2. Furthermore, BREAD exhibits faster convergence as the rank increases, since larger rank offers higher freedom of search directions.

---

> > ### Comment · Reviewer_3qN8 · 2024-11-24
> > **I have raised my score but there are still some concerns**
> >
> > I thank the authors for addressing my questions. I have updated my score to 5; however, several concerns remain:
> > 1. **Limited Technical Contributions.** I still find the technical contributions of BREAD to be limited. The techniques employed are already present in LoRA, GaLore, and LOMO, and they are utilized for similar purposes in this work.
> > 2. **Baselines**
> > - Results for LOMO are still missing.
> > - Quantitative results for other variants (BREAD-partial, BREAD-SGD, BREAD-SGD-partial) are not included in Tables 2 and 3.
> > 3. **Statistical Significance.** To evaluate the robustness of the fine-tuning results, the authors should report standard deviations across multiple random seeds.

---

> ### Author Response · Authors · 2024-11-25
> **Response to your concerns**
>
> Thank you for reading our response carefully. We greatly appreciate your raising of scores. We now address your remaining concerns.
>
> **A. Limited technical contributions.** We do agree that ideas like low-rank update (i.e., Adapter, LoRA, etc) and layer-wise SGD update (i.e., LOMO) are existing techniques for memory efficient finetuning of LLMs. Indeed, many subsequent training approaches accepted at top machine learning conferences have drawn inspiration from these techniques. Our main novelty lies in that
>
> 1. **BREAD is a versatile BCD algorithmic framework rather than tailored to LoRA or LOMO.** The main theme of the manuscript is to accelerate BCD with landscape correction. We observe that the suboptimal landscape of BCD may slow down the convergence.  Then, BREAD is proposed to be *a general accelerated BCD framework* that incorporates a correction step to all / some of the inactive blocks. Importantly, this correction step can utilize *any* lightweight memory efficient finetuning techniques. We currently use LoRA and LOMO, but any memory efficient method called X can be used in BREAD once X is proposed in the future. We will release our code, which enables the use of any memory efficient finetuning method for the correction step in BCD. Thus, we would like to emphasize that BREAD is essentially and fundamentally a BCD method aimed at accelerating the vanilla BCD scheme rather than tailored to LoRA or LOMO. Consequently, the framework retains desirable features of BCD, including high-rank updates, full parameter training, memory efficiency, high-precision updates, and time efficiency. Hence, we believe the proposed general algorithmic framework is both novel and impactful.
>
> 2. **The first study to technically identify the issue of BCD for neural nets.** Technically speaking, we claim that this is *the first* study to theoretically identify the issue in BCD for neural nets training, where inactive blocks that are far from optimality can interfere with the optimization of the active block, as acknowledged by *Reviewer Ajgg*. This theoretical insight serves as the foundation of BREAD and explains why BREAD can accelerates the vanilla BCD scheme to a large extent by simply incorporating any lightweight corrector for the inactive blocks.
>
> 3. **Substantially improved performance is also a non-trivial contribution.** We would like to emphasize the impressive performance of the proposed general algorithmic framework. With minimal memory overhead, it consistently outperforms BAdam across both math benchmarks, MT-bench, and loss convergence.  We believe that large performance improvement should also be recognized as substantial contributions. For example, consider masked attention versus standard attention. Both utilize the same fundamental concepts with small modifications, yet masked attention demonstrates significantly improved performance, making it an important contribution to the LLM community. Similar comparisons can be made between LoRA and Adapter. Thus, we argue that BREAD offers both nontrivial technical contributions and substantial performance enhancements.
>
> We hope these perspectives highlight the nontrivial contribution and novelty of our approach, which we believe go beyond “A+B” combination.
>
> **B. Baseline results.** As requested by reviewer, we have added the math evaluation results for LOMO and BREAD variants in Appendix D.4, Table 3 of the latest manuscript. All the BREAD variants outperform BAdam in 0-shot setting and show more significant improvements over other memory-efficient baselines. The variants with partial update have slight performance degrade compared with their full counterparts.
>
> **C. Statistical significance.** We perform 3 independent trials of finetuning the Llama 3.2-1B on MathInstruct dataset. The averaged math benchmark scores, along with their standard deviations, are reported in Appendix D.4, Table 4 of the latest manuscript. Due to time and resource constraints, we were only able to run experiments on the 1B model. In both the 0-shot and 4-shot settings, BREAD outperforms BAdam across all tasks, supporting the effectiveness of the landscape correction. Results for the 8B model and additional BREAD variants will be included in future versions of the manuscript, given more time.
>
> We thank the reviewer for the constructive feedback. We hope our response resolves your concerns on the novelty of BREAD and can earn your support for acceptance. If there is any additional concern, please feel free to let us know. We are happy to provide any additional clarifications or details.

---

> ### Author Response · Authors · 2024-12-01
> **Looking Forward to Your Feedback**
>
> Thank you for carefully reviewing our response and actively participating in the discussion phase. We hope that our response clarifies the contribution of our work and highlights its non-trivial impact. We have included the results of BREAD variants and tried our best to conduct a preliminary statistical significance study. As the deadline for the discussion phase approaches, we would greatly appreciate any further feedback you may have. If you have any additional questions or concerns, please let us know so that we have enough time to address them.

---

### Official Review · Reviewer_aqKv · 2024-11-04

**Soundness:** 3
**Presentation:** 3
**Contribution:** 3
**Rating:** 6
**Confidence:** 3

**Summary:**

The authors propose integrating BCD with landscape correction to address two issues in fine-tuning LLMs with BCD: 1. inefficient computation due to backpropagation through deeper layers and 2. limited exploration of the optimization landscape inherent to BCD. They provide theoretical insights based on a three-layer neural network and demonstrate empirical performance improvements, albeit with increased computational and memory requirements.

**Strengths:**

1. The proposed method is easy to follow and interesting, involving the addition of correction parameters during BCD to improve performance.
2. The paper presents results on LLaMA models across various tasks to validate the effectiveness of the proposed method.

**Weaknesses:**

1. The algorithm lacks a theoretical convergence rate guarantee, although establishing such a rate for BCD optimization is challenging.
2. In Appendix B.1 of the BAdam paper, various ordering strategies for block partitioning in BAdam are investigated; however, this paper neither provides any rationale nor presents similar results.
3. It is surprising to observe that BREAD-SGD-partial demonstrates convergence similar to BREAD-SGD, and the reason for this behavior is unexplained.
4. I believe there is an error in the example used in Proposition 2. Specifically, the expressions $Ce_1=W_3^{(1)}-y$ and $y-W_3e_1=y-W_3^{(1)}$ does not subtract to 0.

**Questions:**

See the above weaknesses for questions.

---

> ### Author Response · Authors · 2024-11-21
> **Authors' Response**
>
> We appreciate your recognition on the effectiveness of BREAD, and thank you for recommending acceptance with a score of 6. We reply to your comments below. Major changes in the manuscript are highlighted in blue.
>
> **A. Theoretical convergence guarantee.** Under the assumption on function smoothness, bounded derivatives, and deterministic gradient, we can establish the following descent property for BREAD:
> $$ H( W^{t+1}) - H( W^{t}) \leq - \mathcal{O}(\alpha K) || \nabla H( W^t) ||^2. $$
>
> We sketch the proof here and provide the detailed analysis in Appendix C of the latest manuscript. Built upon the analysis of BAdam, we only need to analyze the descent induced by updating correction matrices. At the beginning of block epoch $t$, block sub-problem $i$, let us define $W_{j, i}^{t}$ as the parameter of block $j$, and $W_i^t:=(W_{1,i}^{t+1}, W_{2,i}^{t+1}, \cdots, W_{i,i}^{t+1}, W_{i+1,i}^{t}, W_{D,i}^{t})$ be the parameter of the full model. By the smoothness of the function, the descent of one block sub-problem is given by
> $$
> \begin{align}
> H(W\_i^t) - H(W\_{i-1}^t) &\leq \sum\_{j \in [D] \backslash i} \langle \nabla_{j}H(W\_{i-1}^{t}), W\_{j, i}^{t+1} - W\_{j, i-1}^{t+1} \rangle + \sum_{j \in [D] \backslash i} \frac{L\_j}{2} || W\_{j, i}^{t+1} - W\_{j, i-1}^{t+1} ||^2 \\\\
>     &\quad + \langle \nabla\_{W\_i} H(W\_{i-1}^{t}), W\_{i, i}^{t+1} - W\_{i, i-1}^{t+1} \rangle + \frac{L\_i}{2} || W\_{i, i}^{t+1} - W\_{i, i-1}^{t+1} ||^2.
> \end{align}
> $$
>
> The last two terms represents the descent induced by the active block. The first two terms are induced by the update of correction matrices, which can be shown to be in the order $-\mathcal{O}(K\beta)||\nabla_{W_j}H(W_{i-1}^{t})||^2 + \mathcal{O}(\beta^2 K)$, where $\beta$ is the step size used for correction matrices. Hence, we can control the $\beta$ to ensure the negativity of this term and they contribute to the additional descent compared with BAdam. Consequently, we can establish the descent property and thereby the convergence of BREAD.
>
> **B. Suggestion on block partitioning and other algorithmic setup.** We have conducted more experimental results to study the hyperparameters of BREAD; see Appendix D.1 of the latest revised manuscript.
>
> *Choice of inner Adam steps $K$.* As shown in Figure 4.a, large $K$ leads to faster convergence. In particular, setting $K=512$ achieves almost twice faster convergence as setting $K=128$. Based on this finding, we suggest choosing $K=512$ for fast convergence. We leave more scientific study on $K$ as a future work. We thank reviewer for pointing it out, which helps us establish a deeper understanding on the effect of K.
>
> *Choice of correction matrices' rank $r$.* As shown in Figure 4.b, rank-2 correction can boost the performance significantly, and larger rank will lead to faster convergence. Hence, we suggest to choose $r$ to be a large value that is memory-affordable.
>
> *Partition ordering strategy.* In Figure 4.c, we show that different block ordering strategies does not result in evident performance deviation, and can be chosen freely.
>
> **C. Similar convergence of BREAD-SGD-partial and BREAD-SGD.** We are aware of this phenomenon as well. Compared with BREAD-SGD-partial, the BREAD-SGD will update the layers closer to input (the shallow layers) in a more frequent manner. We conjecture that the input layers have relatively smaller gradient or flatter landscape that is not sensitive to small update, such that the SGD update of these layers does not change the objective evidently. Consequently, BREAD-SGD-partial and BREAD-SGD will exhibit similar convergence. Notably, there is a convergence gap between BREAD and BREAD-partial. This may due to that Adam scales the update of each optimization variables separately, and may induce larger learning rate for the shallow layers. We leave the scientific study on this issue as a future work.
>
> **D. Typo in Proposition 2.** We thank the reviewer for pointing out the typo, where $ C$ should be corrected as $C=\left[ y - W_3^{(1)}, 0, \cdots, 0 \right]$. We have updated the latest manuscript accordingly.

---

> ### Author Response · Authors · 2024-11-25
> **Looking Forward to Your Response**
>
> We hope our response addresses most of your concerns. Specifically, we have provided the convergence analysis of BREAD, demonstrating that it is a descent algorithm. Additionally, we have conducted an ablation study on the ordering strategies, which shows that the choice of ordering does not significantly impact the algorithm’s convergence. We also offer a preliminary discussion on the similar convergence behavior of BREAD-SGD and BREAD-SGD-partial.
>
> As the deadline for the discussion phase approaches, we would greatly appreciate your feedback. Should you have any further questions or comments, please let us know, so we have sufficient time to address them. We would be happy to provide any additional clarifications or details.

---

### Official Review · Reviewer_Ajgg · 2024-11-04

**Soundness:** 3
**Presentation:** 3
**Contribution:** 2
**Rating:** 5
**Confidence:** 4

**Summary:**

Recently, a method called BAdam, which utilizes block coordinate descent (BCD), has gained attention in LLM fine-tuning for its memory efficiency.
This study reveals a drawback of BCD optimization in neural network training: the frozen blocks can narrow the optimization landscape, potentially misleading the training of the active block, resulting in suboptimal solutions.
Based on this intuition, the authors propose the BREAD method, which corrects the loss landscape of BCD. Also, the authors experimentally demonstrate the superiority of BREAD over the existing baselines in terms of both memory-efficiency and performance.

**Strengths:**

1. It is the first study to identify the issue in BCD, where frozen blocks that are far from optimality can interfere with the optimization of the active block.
2. It theoretically demonstrates, through a regression problem on a three-layer shallow network, that suboptimal solutions may indeed arise.
3. For LLM fine-tuning tasks, it shows improvements over existing baselines in both memory efficiency and performance.

**Weaknesses:**

I have several concerns.

1. Although it is intuitive that BCD optimization can lead to suboptimal solutions in a regression problem using a 3-layer shallow network, the situation could differ for a general L-layer deep neural network. Moreover, since LLM fine-tuning often involves classification tasks rather than regression, it would be helpful to have a theoretical analysis or at least some intuition on how this approach would perform with loss functions like cross-entropy.

---

2. Convergence analysis is missing. For a landscape correction scheme like BREAD, there should be at least some convergence results, even in the context of convex optimization, to show whether it is a provably convergent algorithm. I think it is crucial in optimization literature.

---

3. The performance improvement in the main experiment of LLaMA-3 fine-tuning appears marginal. It would be beneficial to include more comprehensive experimental results across a variety of settings, such as with alternative architectures like Phi-3 or Mistral, and other benchmark tasks.

---

4. Additionally, according to Table 1, BREAD is slower than BAdam in terms of Epoch GPU hours. Therefore, for a clearer understanding of the BREAD method, it would be helpful to include a figure comparing learning curves in terms of wall-clock time rather than just the number of iterations, as shown in Figure 3.

**Questions:**

Please refer to the weaknesses.

---

> ### Author Response · Authors · 2024-11-21
> **Authors' Response: Part I**
>
> Thank you for acknowledging our theoretical insight and recognizing the algorithm's performance. We now provide clarifications on the algorithmic design and address your concerns in a point-by-point manner. Major changes in the manuscript are highlighted in blue.
>
> **A. Generalize analysis to $L$-layer model and cross entropy loss.** Similarly to the 3 layer case, the $L$-layer model also suffers from the suboptimal landscape issue for both regression and classification problem. We explain the intuition for the suboptimality, and provide the formal analysis in Appendix C of the revised manuscript. In particular, let us consider an $L$-layer model:
> $$
> \begin{align}
> z_1 &= \sigma(W_1 x) \\\\
> z_{i} &= \sigma(W_i z_{i-1}), \quad i=2, \cdots, L-1 \\\\
> \hat{y} &= W_{L} z_{L-1}
> \end{align}
> $$
> where $\sigma(x) = \max(0, x)$ is the ReLU activation function and $ z_i \in \mathbb{R}^{d_i}$.
>
> *Suboptimality analysis for regression task.* The regression loss is given by $||\hat{y} - y||^2$.
> * **Effect of freezing $W_L$.** When $W_L$ is full column rank, the optimal $z_{L-1}$ we seek to fit is the least square solution $z_{L-1}^* = (W_L^\top W_L)^{-1}  W_L^{\top} y$. When $z_{L-1}^{*}$ contains negative entries, it cannot be fit due to the non-negativity of the ReLU function, which induces the suboptimality:
> $$\min_{W_1, \cdots  W_{L-1}} ||\hat{ y} - y||^2 > \min\_{W_1, \cdots, W_L} ||\hat{y} - y||^2.$$
>
> * **Effect of freezing intermediate layers.** Each intermediate layer performs the transformation $z_{i} = \sigma( W_i  z_{i-1}) := \mathcal{M}_i $. When $W_i$ is trainable, we have range($\mathcal{M}_i$) = $\mathbb{R}^{d_i+}$ when $z\_{i-1} \neq 0$. However, when $W\_i$ is frozen, range($\mathcal{M}\_i$) is limited to the projected "restricted" column space of $W\_i$, where "restricted" means the column combination should be positive, due to the positivity of $z\_{i-1}$.
>
> *Suboptimality analysis for classification task with cross entropy loss.* Without loss of generality, let us assume the ground truth label is the first class. The cross entropy loss is given by $-\log \left( \exp\hat{y}\_1 / (\sum\_{i=1}^{m} \exp \hat{y}\_i) \right)$, where $m$ is the number of classes. Consider the case where the weight of the last layer $W_L$ has a row with the same weight as the first row, i.e. $\exists j$ such that $ W_{L}^{(j)} = W_{L}^{(1)}$, we have $\hat{y}\_j = W\_{L}^{(j)} z\_{L-1} = W\_{L}^{(1)} z\_{L-1} = \hat{y}\_1$. In this case, we will never be able to drive the loss down to $- \log \frac{1}{2}$:
> $$ -\log \left( \exp\hat{y}\_1 / (\sum\_{i=1}^{m} \exp \hat{y}\_i) \right) > -\log \left( \frac{\exp\hat{y}\_1}{\exp\hat{y}\_1 + \exp\hat{y}\_i} \right) = - \log \frac{1}{2}. $$
> While it is not common for $ W_L$ to have two same rows, one can expect large error when there are rows that form small angle, i.e.
> $\frac{\left \langle w_{L}^{(i)}, w_{L}^{(j)} \right \rangle}{||  w_{L}^{(i)}|| ||w_{L}^{(j)}||} \approx 1$.

---

> ### Author Response · Authors · 2024-11-21
> **Authors' Response: Part II**
>
> **B. Convergence analysis of BREAD.** Under the assumption of bounded derivatives, the Lipschitz gradient, and with certain step size conditions on correction matrices, BREAD has the following descent property:
> $$H(W^{t+1}) - H(W^{t}) \leq - \mathcal{O}(\alpha K) || \nabla H(W^t) ||^2.$$
> We sketch the proof here and provide the detailed analysis in Appendix C of the latest manuscript. Built upon the analysis of BAdam, we only need to analyze the descent induced by updating correction matrices. At the beginning of block epoch $t$, block sub-problem $i$, let us define $W_{j, i}^{t}$ as the parameter of block $j$, and $W_i^t:=(W_{1,i}^{t+1}, W_{2,i}^{t+1}, \cdots, W_{i,i}^{t+1}, W_{i+1,i}^{t}, W_{D,i}^{t})$ be the parameter of the full model. By the smoothness of the function, the descent of one block sub-problem is given by
> $$
> \begin{align}
> H(W\_i^t) - H(W\_{i-1}^t) &\leq \sum\_{j \in [D] \backslash i} \langle \nabla_{j}H(W\_{i-1}^{t}), W\_{j, i}^{t+1} - W\_{j, i-1}^{t+1} \rangle + \sum_{j \in [D] \backslash i} \frac{L\_j}{2} || W\_{j, i}^{t+1} - W\_{j, i-1}^{t+1} ||^2 \\\\
>     &\quad + \langle \nabla\_{W\_i} H(W\_{i-1}^{t}), W\_{i, i}^{t+1} - W\_{i, i-1}^{t+1} \rangle + \frac{L\_i}{2} || W\_{i, i}^{t+1} - W\_{i, i-1}^{t+1} ||^2.
> \end{align}
> $$
>
> The last two terms represent the descent induced by the active block. The first two terms are induced by the update of correction matrices, which can be shown to be in the order $-\mathcal{O}(K\beta)||\nabla_{W_j}H(W_{i-1}^{t})||^2 + \mathcal{O}(\beta^2 K)$, where $\beta$ is the step size used for correction matrices. Hence, we can control the $\beta$ to ensure the negativity of this term and they contribute to the additional descent compared with BAdam. Consequently, we can establish the descent property and thereby the convergence of BREAD.
>
> **C. "The performance improvement in the main experiment of LLaMA-3 fine-tuning appears marginal. It would be beneficial to include more comprehensive experimental results"**. We argue that the performance improvements on Llama 3 models are actually significant, where BREAD even outperforms Adam in averaged score for 8B model's 0-shot setting. Furthermore, we have carefully tuned the hyperparameter of baseline methods. For instance, the setting of LoRA rank $r=64$ and $\alpha=2r$ in our paper are shown to be a preferable choice, based on the comprehensive study in [1]. Due to the time-consuming nature of LLM experiments, we are unable to provide a comprehensive evaluation on architectures such as Mistral and Phi-3, which we leave as a future work. We remark that our paper has included additional experiments on the 70B model to demonstrate the scalability of the proposed methods.
>
> **D. Convergence in terms of time.** We have added the convergence to compare BAdam and BREAD variants in terms of time; see Appendix D.2 of the latest manuscript. In terms of wall-clock time, BREAD-SGD-partial achieves the fastest convergence among all the variants and surpasses BAdam by a large margin. BREAD-SGD and BREAD-partial achieves comparable convergence speed as BAdam. Notably, All the BREAD variants achieve lower training loss than BAdam at the end of training.
>
> We believe our rebuttal has addressed all of your concerns with our best efforts. We hope it earns your trust in the quality of our work and your support for acceptance.
>
> **Reference**
>
> [1] LoRA vs Full Fine-tuning: An Illusion of Equivalence

---

> ### Author Response · Authors · 2024-11-25
> **Looking Forward to Your Response**
>
> We hope that our response addresses most of your concerns. Specifically, we have shown that suboptimal landscape issue exists in general $L$-layer neural network and cross-entropy loss. As requested by reviewer, we also included the convergence analysis and provided the loss convergence in terms of time.
>
> Since the deadline of the discussion phase is approaching, we would highly appreciate to receive feedback from you. If you have any further questions or remarks, please let us know so that we will have enough time to address your concerns. We will be more than happy to provide additional clarifications and details.

---

> ### Author Response · Authors · 2024-11-26
> **Numerical Verification on Cross Entropy Loss and L-layer Model**
>
> Thank you for carefully reviewing our response. To further support our suboptimality analysis in $L$-layer neural network training and cross entropy loss, we have conducted additional experiments, which are detailed in Appendix D.4 of the latest manuscript.
>
> Specifically, our experiment shows that when optimizing the cross entropy loss, BREAD with rank-1 correction significantly accelerates block coordinate descent (BCD), which is aligned with our analysis on classification problem. Notably, BAdam only begins to converge rapidly after the final layer has been trained. This phenomenon is consistent with our theoretical insight that a rank-deficient final layer may hinder convergence. We also trained an 8-layer neural network and observed similar acceleration for BREAD, which corroborates our suboptimality analysis of L-layer model.
>
> We hope our experimental results, along with the theoretical analysis in previous response, address your concerns and highlight the novelty and effectiveness of our algorithmic design. If there is any further concern or question, please feel free to let us know. We would be happy to provide additional clarifications and details.

---

> ### Author Response · Authors · 2024-12-01
> **Looking Forward to Your Feedback**
>
> Thank you for reviewing our paper. We believe we have addressed all the concerns raised in your initial review, and have made significant improvements to the manuscript based on your feedback. As the deadline for the discussion phase approaches, we would greatly appreciate your first-round response. If you have any further questions or comments, please let us know so that we have sufficient time to address them.

---

> > ### Comment · Reviewer_Ajgg · 2024-12-03
> > **Response to Authors**
> >
> > I appreciate the authors' response.
> >
> > I think that the authors successfully address the most of my concerns. So, I decided to raise the score up to 5.
> > However, I recommend that the authors should elaborate more on the intuition and theory for general deep networks.

---

> ### Author Response · Authors · 2024-12-03
> **Response to Your Further Suggestion**
>
> Thank you so much for raising the score. We are happy to know that most of your concerns have been successfully addressed. We appreciate your suggestion, and we now elaborate BCD's suboptimality issue for general deep network from both intuition and theory perspectives.
>
> * **Suboptimality of general $L$-layer network**
>   * **Intuition**: **1)** Freezing the last layer restricting the output space from covering the optimal solution. **2)** Freezing the intermediate layers requires the model to learn from potential poor features.
>   * **Theory**: We establish the condition where BCD's suboptimality exists for general $L$-layer network, i.e. $(W_{L}^{\top}W_{L})^{-1}W_L^{\top}y$ contains at least one negative entry, which holds with high probability under the widely adopted Gaussian initialization. Please also see our Response A to reviewer KSei for justification.
>   * **Numerical verification**: In Appendix D.4 Figure 7, we show that by correcting the optimization landscape, BREAD converges significantly faster than BCD for 8 layer neural network training.
> * **Suboptimality for classification problem**
>   * **Intuition**: When the last layer's weight contains two rows that form small angle, the corresponding classes' probability will be close to each other. Suppose one of the class is the groundtruth, it is hard to increase the probability of the groundtruth class while decreasing the probability of the other.
>   * **Theory**: We show that the loss can never be lower than $-\log(1/2)$ when there exists two identical the rows in last layer. We will include additional analysis for the case where two rows form small angles in our manuscript when revising is allowed.
>   * **Numerical verification**: In Appendix D.4 Figure 6, we show that when using cross entropy loss, BCD exhibits poor convergence before training the last layer. In comparison, BREAD converges significantly faster than BCD, due to its effective landscape correction.
>
> We hope that our response is satisfactory to you. Our rebuttal takes your comments into full consideration, and we hope it will earn your support for acceptance.

---

### Author Response · Authors · 2024-11-22
**General Response**

We sincerely thank all the reviewers for their constructive feedback and for their acknowledgement of **novelty of our analysis on BCD's potential suboptimality** (Reviewer Ajgg, 3qN8), **the innovative algorithmic design** (Reviewer KSei) , and **the superior performance** (all reviewers).

We have polished the manuscript, tried our best to complete all the experiments requested by reviewers, and made clarifications in the revised version. Below, we would like to summarize the major concerns raised by reviewers, as well as our response in addressing their concerns.

* **Novelty of the algorithm.** We emphasize that BREAD is a well-motivated algorithmic framework rather than a trivial combination of BCD and LoRA/LOMO, and it naturally incorporates any light-weight optimizers for alleviating suboptimality issue and accelerates the convergence of BCD. We have justified the effectiveness of BREAD framework through theoretical insights (suboptimality landscape issue and convergence), computational-efficient design (marginal increase of memory cost compared to BCD), and competitive downstream performance under limited memory budget (outperforms memory-efficient baselines in math and instruction tuning tasks).

* **Theoretical convergence of BREAD.** We have provided the convergence result of BREAD; see Appendix C of the manuscript for the detailed analysis. In particular, under the assumptions of bounded derivatives, Lipschitz gradient, we have shown that BREAD with deterministic gradient is a descent method and can find $\varepsilon$-stationary point with $\mathcal{O}(\varepsilon^{-2})$ gradient evaluations.

* **Ablation study on hyperparameters.** We have conducted ablation study on the Adam inner step $K$, rank of the correction matrices $r$, and block ordering strategies; see Appendix D.1 of the manuscript for the detailed results. Experiments show that increasing $K$ or $r$ consistently accelerates the convergence of BREAD for the examined range, and different ordering strategies exhibit similar convergence.

* **Convergence in terms of time.** We have updated the training loss convergence in terms of wall-clock time in Appendix D.2. BREAD-SGD-partial achieves the fastest convergence in terms of time, and BREAD-SGD and BREAD-partial surpasses BAdam at certain points. All the BREAD variants achieve lower training loss than BAdam after 3 epochs.

We hope that these revisions and responses are satisfactory to all reviewers. Once again, we truly appreciate the very detailed and constructive feedback, which has helped us greatly in improving the quality of the manuscript.

---

### Meta-Review · Area_Chair_3pfS · 2024-12-18

**Metareview:**

The paper presents the BREAD framework, aimed at addressing inefficiencies in the block coordinate descent (BCD) method for fine-tuning large language models (LLMs). It proposes an innovative approach to landscape correction that seeks to enhance convergence and performance while maintaining a lightweight optimization process. This research carries significant implications for improving the performance of LLMs in various applications.

Reviewer opinions on the paper varied, with some acknowledging the potential of the experimental results while expressing concerns about the novelty of the technical contributions. While one reviewer recognized the intriguing nature of the issues addressed by BREAD, others felt that it merely combined existing techniques like LoRA and BCD without introducing substantial innovations. There was no unanimous agreement among the reviewers, indicating a lack of consensus on the paper's overall impact.
Following the authors' rebuttal, several concerns remained unresolved. Key issues included the perceived limitations in the technical contributions, as reviewers noted that the concepts presented seemed to lack novelty. Additionally, specific requests for further experimental validation and clearer exposition of the methodology were not fully addressed. Despite efforts by the authors to enhance clarity and provide additional data, the fundamental apprehensions regarding the paper's originality and contributions persisted.
In light of the reviewers' consistent negative feedback and the authors' inability to fully alleviate concerns despite the rebuttal, I recommend rejecting this submission. Overall, the paper does not meet the necessary standards for acceptance at this conference.

**Additional Comments On Reviewer Discussion:**

Reviewer opinions on the paper varied, with some acknowledging the potential of the experimental results while expressing concerns about the novelty of the technical contributions. While one reviewer recognized the intriguing nature of the issues addressed by BREAD, others felt that it merely combined existing techniques like LoRA and BCD without introducing substantial innovations. There was no unanimous agreement among the reviewers, indicating a lack of consensus on the paper's overall impact.
Following the authors' rebuttal, several concerns remained unresolved. Key issues included the perceived limitations in the technical contributions, as reviewers noted that the concepts presented seemed to lack novelty. Additionally, specific requests for further experimental validation and clearer exposition of the methodology were not fully addressed. Despite efforts by the authors to enhance clarity and provide additional data, the fundamental apprehensions regarding the paper's originality and contributions persisted.

---

### Decision · Program_Chairs · 2025-01-22

Reject